

**Development of a Universal Correction Algorithm for Filter-Based Absorption Photometers**
Hanyang Li[1], Gavin R. McMeeking[2], and Andrew A. May[1]
1 The Ohio State University Department of Civil, Environmental, and Geodetic Engineering,
Columbus, Ohio, USA
2 Handix Scientific, LLC, Boulder, Colorado, USA
*Corresponding author*: Andrew A. May (may.561@osu.edu).
Abstract
Among the various measurement approaches to quantify light absorption coefficient ($B_{abs}$), filter-
based absorption photometers are dominant in monitoring networks around the globe. Numerous
correction algorithms have been introduced to minimize the artifacts due to the presence of the
filter in these instruments. However, from our recent studies conducted during the Fire Influence
on Regional and Global Environments Experiment (FIREX) laboratory campaign, corrected filter-
based $B_{abs}$ remains biased high by roughly a factor of 2.5 when compared to a reference value
using a photoacoustic instrument for biomass burning emissions. Similar over-estimations of $B_{abs}$
from filter-based instruments exist when implementing the algorithms on six months of ambient
data from the Department of Energy (DOE) Atmospheric Radiation Measurement (ARM)
Southern Great Plains (SGP) user facility from 2013 (factor of roughly 3). In both datasets, we
observed an apparent dependency on single scattering albedo (SSA) and absorption Ångström
exponent (AAE) in the agreement between $B_{abs}$ based on existing correction factors and the
reference $B_{abs}$. Consequently, we developed a new correction approach that is applicable to any
filter-based absorption photometer that includes light transmission from the filter-based instrument
as well as the derived AAE and SSA. For the FIREX and SGP datasets, our algorithm results in
good agreement between all corrected filter-based $B_{abs}$ values from different filter-based
instruments and the reference (slopes ≈ 1 and $R^2$ ≈ 0.98 for biomass burning aerosols and slopes
≈ 1.05 and $R^2$ ≈ 0.65 for ambient aerosols). Moreover, for both the corrected $B_{abs}$ and the derived
optical properties (SSA and AAE), our new algorithms work better or at least as well as the two
common PSAP-based correction algorithms. The uncertainty of the new correction algorithm is
estimated to be ~10%, considering the measurement uncertainties of the operated instruments.
Therefore, our correction algorithm is universally applicable to any filter-based absorption
photometer and has the potential to "standardize" reported results across any filter-based
instrument.
1. Introduction
Light-absorbing atmospheric aerosols directly affect the Earth's energy budget by absorbing solar
radiation, leading to a warming effect when they are suspended in the atmosphere and to the
melting of snow and ice following deposition (Bond and Bergstrom, 2006; Boucher, 2015; Horvath,
1993). For decades, scientists have conducted field experiments around the globe to investigate
how absorbing aerosols influence the atmospheric radiative balance and interact with clouds (e.g.,
Andrews et al. (2011); Cappa et al. (2016); Lack et al. (2008b); Rajesh and Ramachandran (2018);
Schwarz et al. (2008)). These experiments may be performed at fixed stations (e.g., observation
sites maintained by the Department of Energy (DOE) Atmospheric Radiation Measurement (ARM)
program or the National Oceanic and Atmospheric Administration (NOAA) Global Monitoring



Division (GMD)) or on mobile platforms (e.g., car trailer, aircraft, and ship), typically involving
the measurements of aerosol chemical, physical, and optical properties. Crucial to the
quantification of the radiative forcing of absorbing aerosols are measurements of the absorption
coefficient ($B_{abs}$). For example, long-term monitoring of $B_{abs}$ provides essential data to evaluate
chemistry-climate model simulations (e.g., Chen et al. (2019); Vignati et al. (2010)), while
intensive measurements of $B_{abs}$ during short-term field campaigns allow for the investigation of
optical properties that govern features of aerosol forcing (e.g., McMeeking et al. (2014); Olson et
al. (2015)).
A variety of instruments have been used to measure $B_{abs}$, which generally classified into two large
categories: filter-based techniques and photoacoustic techniques (Lack et al., 2014; Moosmüller
et al., 2009). The major difference between the two categories of technique is that $B_{abs}$ is measured
after the aerosols are deposited on the filter media in the filter-based instruments, while the aerosols
are characterized within an air stream in the photoacoustic instruments. Compared to the filter-
based instruments, the photoacoustic instruments have the advantage of avoiding potential artifacts
due to the contact of aerosols with filters; therefore, they are often used as the reference instruments
in inter-comparison studies of aerosol absorption (e.g., Arnott et al. (2005); Davies et al. (2019);
Jiang et al. (2018); Li et al. (2019); Schmid et al. (2006); Sheridan et al. (2005)).
Filter-based absorption photometers have been widely used at observational sites around the world
due to their ease of operation and relatively low cost. Numerous instruments can be classified as
filter-based absorption photometers including the Radiance Research Particle Soot Absorption
Photometer (PSAP), the NOAA Continuous Light Absorption Photometer (CLAP), the Brechtel
Manufacturing Tricolor Absorption Photometer (TAP), the Magee Scientific Aethalometer
(AETH), and the Thermo Scientific Multi-Angle Absorption Photometer (MAAP). Operationally,
all of these instruments are similar in that aerosols are deposited onto a filter and the reduction in
the transmission (Tr) of light by the particles (sometimes called attenuation (ATN)) is used to infer
$B_{abs}$. Where the instruments may differ is that some are multi-wavelength (multi-$\lambda$) instruments
(e.g., 3$\lambda$-PSAP, CLAP, TAP, 7$\lambda$-AETH models), while others are not (e.g., 1$\lambda$-PSAP, other AETH
models, MAAP).
One challenge with filter-based absorption photometers is that biases can arise due to the presence
of the filter. For example, light scattering by particles loaded onto the filter or by the filter itself
may affect the transmission of light (e.g., (Arnott et al., 2005; Bond et al., 1999)); non-absorbing
material may result in absorption enhancement (e.g., (Cappa et al., 2008)); or organic vapors
adsorbed to the filter may itself absorb light (e.g., (Subramanian et al., 2007)). Consequently,
various correction algorithms exist to minimize these biases, but they are often specific only to
certain instruments. For example, some are applicable to the PSAP, CLAP, and TAP (e.g., (Bond
et al., 1999; Müller et al., 2014; Ogren, 2010; Virkkula, 2010; Virkkula et al., 2005)), while others
are applicable to the AETH (e.g., (Arnott et al., 2005; Collaud Coen et al., 2010; Drinovec et al.,
2017; Kirchstetter and Novakov, 2007; Schmid et al., 2006; Virkkula et al., 2007, 2015;
Weingartner et al., 2003)).
Although the equations associated with these existing correction algorithms are different, they
share some commonalities. For example, the filter-based absorption photometers are assessed
using laboratory (e.g., ammonium sulfate, fullerene soot) or ambient aerosols during experiments
which include reference measurements of $B_{abs}$. These reference measurements often include either
direct photoacoustic $B_{abs}$ or inferred $B_{abs}$ as the difference between the extinction coefficient ($B_{ext}$)



and the scattering coefficient ($B_{scat}$). Correction equations are developed by comparing data between the filter-based instrument and the reference instrument, where the equations often contain one term that accounts for filter loading effects and another that accounts for multiple-scattering effects. Consequently, the correction equations frequently incorporate both Tr and either $B_{scat}$ or the single-scattering albedo (SSA) to account for these effects. However, even when the correction algorithms are applied, potential issues can remain such as:

1. Corrected filter-based $B_{abs}$ may remain biased high relative to a reference value of $B_{abs}$ (e.g., (Arnott et al., 2003; Davies et al., 2019; Lack et al., 2008a; Li et al., 2019; Müller et al., 2011a)).
2. Comparisons between the reference instrument and $B_{abs}$ corrected by different algorithms can yield variable agreement (e.g., (Collaud Coen et al., 2010; Davies et al., 2019; Saturno et al., 2017)).
3. Corrected $B_{abs}$ from different filter-based absorption photometers may not agree (e.g., (Davies et al., 2019; Müller et al., 2011a)).
4. Derived products (such as absorption Ångström exponents (AAE)) may differ based on the implemented correction algorithm (e.g., (Backman et al., 2014; Davies et al., 2019)).
5. The agreement between measurements of $B_{abs}$ and estimates of $B_{abs}$ by chemistry-climate models may vary based on the implemented correction algorithm (e.g., (Alvarado et al., 2016)).

The first three issues in this list may arise due to differences in aerosol optical properties between those used in deriving the correction equation and those associated with a given aerosol sample, and these issues can propagate through to the fourth issue. The final issue is arguably most important because evaluation of chemistry-climate models may be severely affected by the differences between different correction algorithms, which may inhibit the modeling community from providing accurate projections of future temperature and precipitation response.

In this work, we seek to address some of these issues. First, we evaluate the CLAP, TAP, and PSAP using two common PSAP-based correction algorithms, namely Bond et al. (1999) as updated by Ogren (2010) and Virkkula et al. (2005) as updated by Virkkula (2010). For brevity, we refer to these corrections as "B1999" and "V2005" for Bond et al. (1999) and Virkkula et al. (2005), respectively, incorporating their respective updates. In addition, we propose "universal" correction algorithms that are applicable to any filter-based absorption photometer (e.g., CLAP, TAP, PSAP, and AETH) across multiple wavelengths by combining observed filter-based $B_{abs}$ with $B_{scat}$ (e.g., from a co-located nephelometer (NEPH)) and reference $B_{abs}$ (e.g., from a co-located photoacoustic instrument). However, in reality (e.g., at long-term observatories), reference values of $B_{abs}$ are rare, and in some cases, complementary $B_{scat}$ measurements may not exist; consequently, we also provide methods to correct filter-based $B_{abs}$ data in these scenarios. To our knowledge, this is the first study to simultaneously evaluate B1999 and V2005 corrections on PSAP "successors" (i.e., CLAP and TAP) and to present a correction algorithm that is broadly applicable to any filter-based absorption photometer. Regarding the latter, even if our correction algorithm has its own limitations, its use can nevertheless standardize the reporting of $B_{abs}$ in long-term datasets.

2. Methodology

We developed the general form for our correction algorithms using CLAP and TAP measurements collected from biomass burning (65 fires in total) during the Fire Influence on Regional to Global Environments Experiment (FIREX) laboratory campaign in 2016. By using biomass burning



emissions, we considered a dataset spanning a broader range of aerosol optical properties (SSA at
652 nm: 0.14-0.98; AAE: 1.25-4.73) than has traditionally been used in developing these
correction algorithms. We then conducted further evaluation and validation of the model using
ambient data, specifically using CLAP measurements from the DOE ARM Southern Great Plains
(SGP) user facility in Lamont, OK, USA (02/01/13 to 07/09/13). Our algorithms were then
extended to the AETH data from the FIREX laboratory campaign and the PSAP data collected at
the SGP site to verify the "universal" nature of the algorithms.
2.1. The FIREX campaign
2.1.1.  Experimental setup
In October and November of 2016, we participated in the laboratory portion of the FIREX
campaign to investigate the wildfire smoke and their impact on the atmosphere. During the
campaign, over 100 burns took place at the U.S. Forest Service's combustion facility at the Fire
Sciences Laboratory (FSL). The fuels burned in this study are representative of western US
ecosystems, such as spruce, fir, various pines, and "chaparral" biome (e.g., manzanita, chamise).
(see Koss et al. (2018) and Selimovic et al. (2018) for more details).
A typical burn lasted for 1-3 hours depending on the smoke sampling strategies (e.g., stack burns
versus room burns). During each burn, one or multiple "snapshots" of smoke (typical $B_{abs}$ at 652
nm ranged from 100 to 1200 Mm$^{-1}$) were transferred from the combustion room at FSL into a
mixing chamber (210 L) through a long transfer duct (30 m in length, 8" in diameter). The smoke
was then diluted by filter air (~230 LPM) in the chamber. Once the concentration in the chamber
was stable (detected by the Photoacoustic Extinctiometer (PAX) which was operated continuously
through all fires), the smoke was passed to a suite of instruments to obtain aerosol and gas phase
parameters. This chamber also served as an intermediate between the transfer duct and the
instrumentation to minimize potential biases that arose due to different sample flow rates and
sample locations of the instruments. A more detailed description of our experiments can be found
in Li et al. (2019).
2.1.2.  Measurements of aerosol optical properties
During the campaign, five instruments provided measurements of $B_{abs}$ (CLAP, NOAA GMD; TAP,
Brechtel Manufacturing Inc. (BMI); Aethalometer (Model AETH-31), Magee Scientific; and two
PAXs (Model PAX-870 and PAX-405), Droplet Measurement Technologies) and two instruments
provided measurements of $B_{scat}$ (PAX-870 and PAX-405). The instruments included in the present
work are summarized in Table 1.
Both CLAP and TAP provide $B_{abs}$ measurements of the particles deposited on a filter, similar to
PSAP. Different from PSAP, there are multiple filter spots (8 sample spots and 2 reference spots)
cycling of one filter in CLAP and TAP, enabling the instruments to run continuously through two
or three burns without changing filter. In the CLAP and TAP, sample illumination is provided by
LEDs operated at three wavelengths (467, 528, and 652 nm). Here, we apply both B1999 and
V2005 to CLAP and TAP data, similar to previous work (e.g., (Backman et al., 2014; Davies et
al., 2019)).
The key differences between the CLAP and TAP during the FIREX campaign include:



1. The spot change of the CLAP was manually performed when Tr reached approximately 0.5 (or ATN decreased to ~69), while the TAP advanced to a new spot automatically with a Tr threshold set to be 0.5.
2. The spot area, flow rate, and LED-detected wavelengths differed slightly (Table 1).
3. The CLAP recorded $B_{abs}$ every one minute, while the TAP recorded $B_{abs}$ every ten seconds. To enable the following analysis, we compute the 1-minute averages of TAP-derived parameters.
4. For the first portion of the campaign (the first 17 days of the 45-day campaign), Pallflex E70-2075S filters were used in the CLAP while Azumi filters (model 371M, Azumi Filter Paper Co., Japan) were used in the second portion of the campaign (due to a lack of availability of the Pallflex filters). The TAP was equipped exclusively with the Azumi filters throughout the campaign. We apply the filter correction recommended in Ogren et al. (2017) to the CLAP and convert from Pallflex to Azumi filters.
5. BMI substantially re-engineered the CLAP in their development of the TAP.

These differences resulted in variable agreement between the CLAP and TAP during FIREX; however, the two instruments did largely agree within experimental uncertainty (e.g., see Fig. S8 and Fig. S13 in Li et al. (2019)).

A PAX measures $B_{abs}$ and $B_{scat}$ simultaneously for suspended particles using a modulated diode laser. We use these photoacoustic absorption measurements as the reference to evaluate the filter-based $B_{abs}$ and develop our correction algorithms. To enable the evaluation of CLAP and TAP which operate at different wavelengths than the PAXs, we interpolate the measurements of $B_{abs}$ and $B_{scat}$ to the wavelengths of 467, 528, and 652 nm using the values of AAE and scattering Ångström exponents (SAE), similar to Backman et al. (2014) and Virkkula et al. (2005). Theoretically, AAE and SAE fit absorption and scattering as power law functions of wavelength (Bergstrom et al., 2007).

Due to the numerous correction algorithms for the Aethalometer (e.g. (Arnott et al., 2005; Collaud Coen et al., 2010; Kirchstetter and Novakov, 2007; Saturno et al., 2017; Schmid et al., 2006; Virkkula et al., 2007; Weingartner et al., 2003)), we do not evaluate these in the present work to limit the scope. In fact, the majority of our focus is the B1999 and V2005 corrections to TAP and CLAP. However, we still test the performance of the new algorithms on the AETH to explore its applicability to that instrument.

2.2. Measurements of aerosol optical properties at the SGP observatory

The ambient data used in this manuscript are the ground-based aerosol data measured at the SGP observatory from 02/01/13 to 07/09/13 (archived at https://www.archive.arm.gov/discovery/). For evaluation purposes, we randomly selected a range of dates during which the observations are valid (without incorrect, suspect, and missing data). This time period was also subsequent to an upgrade to the 532 nm laser in the three-wavelength photoacoustic soot spectrometer (PASS-3).

At the site, an impactor was used to switch the sampling between two cutoffs (particle diameter <10 μm (PM10) in the first 30 minutes of each hour and <1 μm (PM1) in the latter 30 minutes of each hour). The aerosols exiting from the impactor were dried to RH less than 40% and passed to a CLAP, a PSAP, and two NEPHs. The PASS-3 operated at the site and also provides $B_{abs}$ and $B_{scat}$ for aerosols, but these samples did not pass through the impactor (e.g., characterizing total suspended particles (TSP)). Typical $B_{abs}$ and $B_{scat}$ reported at the site ranged from 0 to 10 Mm$^{-1}$ and 0 to 50 Mm$^{-1}$ at 550 nm, respectively (e.g., (Sherman et al., 2015)). Although the site is rural





(clean background air), long-term transport aerosols (such as mineral dust, absorbing organic
aerosols, and secondary organic aerosols (SOA)) may affect the local aerosol properties (Andrews
et al., 2019).
We preprocess the SGP data in three steps. First, due to the systematic difference of aerosol sizes
between PASS-derived and filter-based absorption, we only include the PM10 observations,
inherently assuming that any differences in the optical properties of PM10 and TSP are negligible.
Then, we smooth the 1-second data into 10-minute averages. Thirdly, we estimate the detection
limits at each of the three wavelengths in the PASS-3 using the data measured during the
"background zero" periods (Allan, 1966) and discard the observations which are below the
detection limits. With a 10-min-averaging-time, the detection limits ($3\sigma$) for the PASS-3 are 0.78
$Mm^{-1}$ (405 nm), 2.01 $Mm^{-1}$ (532 nm), and 0.30 $Mm^{-1}$ (781 nm). For the filter-based instruments,
the detection limits are based on previous studies (See Table 1). Moreover, we only retain the
observations that satisfy $B_{abs}$ (405 nm) > $B_{abs}$ (532 nm) > $B_{abs}$ (781 nm) (or AAE>0), similar to
Fischer and Smith (2018). As with the PAX data from the laboratory, we adjust the PASS-derived
$B_{abs}$ to 467, 528, and 652 nm using the inferred AAE values for each 10-minute average.



**Table 1** Summary of specifications for instruments relevant to this work.

| Instrument | Flow rate (LPM) | Spot area (cm²) | Type of filter | Measured parameters | Response time | Measurement uncertainty | Detection limit (3σ, Mm⁻¹) |
|---|---|---|---|---|---|---|---|
| PAX-870 | 1.0 | - | - | $B_{abs}$ and $B_{scat}$ (870 nm) | 1s | ~11% ($B_{abs}$) ~17% ($B_{scat}$) (Nakayama et al., 2015) | 0.47 ($B_{abs}$) 0.66 ($B_{scat}$) [a] |
| PAX-405 | 1.0 | - | - | $B_{abs}$ and $B_{scat}$ (405 nm) | 1s | 4% ($B_{abs}$) 7% ($B_{scat}$) (Nakayama et al., 2015) | 0.27 ($B_{abs}$) 0.60 ($B_{scat}$) [a] |
| PASS-3 [b] | 1.0 | - | - | $B_{abs}$ and $B_{scat}$ (405, 532, and 781 nm) | 1s | 4 %, 8 %, and 11 % ($B_{abs}$) (Nakayama et al., 2015) | 0.78 (405 nm) 2.01 (532 nm) 0.30 (781 nm) [a] |
| NEPH [b] | 7.5 | - | - | $B_{scat}$ (450, 550, and 700 nm) | 1s | 10% (Anderson et al., 1996) | 0.29 (450 nm) 0.11 (550 nm) 0.17 (700 nm) (5-min average) (Müller et al., 2011b) |
| CLAP | 0.83 ± 0.02 (FIREX) 0.945 (SGP) | 0.199 (FIREX) 0.195 (SGP) | Pallflex E70-2075S and Azumi filter (model 371M) [c] | $B_{ATN}$ and Tr [d] (467, 529, and 653 nm) | 60s | 30% (Ogren et al., 2017) | 0.6 (1-min average), 0.12 (10-min average) (Ogren et al., 2017) |
| TAP | 1.26 ± 0.01 | 0.253 | Azumi filter (model 371M) [c] | $B_{ATN}$ and Tr [d] (467, 528, and 652 nm) | 10s | 30% (Laing et al., 2016) | 2.67 (467 nm) 4.11 (528 nm) 2.13 (652 nm) (30-s average) (Davies et al., 2019) |
| AETH | 2.4 | 0.5 | quartz fiber sampling tape | $B_{ATN}$ and Tr (370, 470, 520, 590, 660, 880, and 950 nm) | 120s | 10% (Sedlacek, 2016) | 0.1 (Sedlacek, 2016) |
| PSAP | 1.0 | 0.178 | Pallflex E70-2075W | $B_{ATN}$ and Tr [d] (470, 522, and 660 nm) | 60s | ~15% (Bond et al., 1999) | 0.3 (Springston, 2016) |

[a] The detection limits of PAX and PASS-3 are determined by Allan deviation analysis (Allan, 1966) of $B_{abs}$ during "background zero".
[b] During the analysis of the data collected at the SGP, we use $B_{abs}$ derived by the PASS and $B_{scat}$ derived by the NEPH to yield the coefficients in the algorithms.
[c] Two types of filters were used during the FIREX campaign (See Sect. 2.1.2).
[d] The operating wavelengths of CLAP, TAP, and PSAP are stated slightly different by the instrument manufactures. We simply use 467, 528, and 652 nm throughout
this manuscript.





**Table 2** Overview of the studies of B1999 and V2005 and the description of our experiments.

| Study | Aerosol source | SSA subset | Range of $B_{abs}$ (Mm$^{-1}$) | Filter-based instrument for $B_{abs}$ | Reference instrument for $B_{abs}$ [a] | Instrument for $B_{scat}$ [a] | Coefficient values in the correction algorithm [b, c] | | |
|---|---|---|---|---|---|---|---|---|---|
| The study in **B1999** | Lab-generated aerosols, including various mixtures of nigrosin and ammonium sulfate | 0.5-1 (550 nm) | 0-800 (550 nm) | One-λ PSAP (550 nm) | The difference between extinction (OEC) and scattering coefficient (NEPH) [d] | NEPH (450, 550, and 700 nm) | $C_1$ = 0.016 ± 0.023 (550 nm) $C_2$ = 1.55 ± 0.25 (550 nm) $C_3$ = 1.02 ± 0.17 (550 nm) | | |
| The laboratory study in **V2005** | Lab-generated aerosols, including various mixtures of kerosene soot, ammonium sulfate, and polystyrene latex | 0.2-0.9 (530 nm) | 0-800 (530 nm) | One-λ PSAP (550 nm), three-λ PSAP (467, 530, 660 nm) | The average of the PA (532 nm and 1064 nm) and the difference between extinction (OEC) and scattering coefficient (NEPH) [d] | NEPH (450, 550, and 700 nm) | 467 nm $C_1$ = 0.015 $C_4$ = 0.377 ± 0.013 $C_5$ = -0.640 ± 0.007 $C_6$ = 1.16 ± 0.05 $C_7$ = -0.63 ± 0.09 | 530 nm $C_1$ = 0.017 $C_4$ = 0.358 ± 0.011 $C_5$ = -0.640 ± 0.007 $C_6$ = 1.17 ± 0.03 $C_7$ = -0.71 ± 0.05 | 660 nm $C_1$ = 0.022 $C_4$ = 0.352 ± 0.013 $C_5$ = -0.674 ± 0.006 $C_6$ = 1.14 ± 0.11 $C_7$ = -0.72 ± 0.16 |
| The ambient study in **V2005** | Ambient aerosols measured during RAOS and NEAQS [e] | 0.75-1 (530 nm) | 0-15 (530 nm) | One- λ PSAP (550 nm), three-λ PSAP (467, 530, 660 nm) | PA (532 nm and 1064 nm) | NEPH (450, 550, and 700 nm) | | | |
| FIREX | Biomass burning aerosols under relatively controlled laboratory conditions | 0.2 -1 (550 nm) | 38-1800 (550 nm) | CLAP (467, 529, 652 nm), TAP (467, 528, 653 nm), AETH (370, 470, 520, 590, 660, 880, 950 nm) | PAX (405 nm and 870 nm) | PAX (405 nm and 870 nm) | See Table 4 and Tables S6-S10 | | |
| SGP (02/01/13 to 07/09/13) | Ambient aerosols collected at the SGP user facility in Lamont, OK | 0.75-1 (530 nm) | 0-8 (550 nm) | CLAP (461, 522, 653 nm), PSAP (470, 522, 660 nm) | PASS (405, 532, and 781 nm) | NEPH (450, 550, and 700 nm) | | | |

[a] The operating wavelengths are based on the manufacturer specifications.
[b] The coefficients provided in Table 2 are the values presented in Ogren (2010) and Virkkula (2010), which are updated from Bond et al. (1999) and Virkkula et
al. (2005), respectively.
[c] We reformulate the correction equations in the original publications to agree with Eq. (4)–(6) in this manuscript. $C_1$ to $C_7$ are the coefficients in the present work.
[d] OEC is optical extinction cell and PA is the instrument using photoacoustic technique.
[e] RAOS and NEAQS are Reno Aerosol Optics Study and New England Air Quality Study, respectively.





2.3. The correction algorithms
In filter-based instruments, the light intensities transmitted through the sample spot and blank spot
of the filter are recorded as $I_s$ and $I_b$, respectively. The logarithmic ratio of the two intensities at
time t is defined as ATN using the Beer-Lambert law:
$$ATN(t) = -100 \times ln\frac{I_s(t)}{I_b(t)} \qquad\qquad 1$$

where ATN = 0 when beginning a new filter spot (t = 0).
The ATN can be related to Tr by normalizing $I_s/I_b$ at time t relative to $I_s/I_b$ at the start of a new
filter spot (t = 0):
$$Tr(t) = \frac{I_s(t)/I_b(t)}{I_s(0)/I_b(0)} = exp(\frac{-ATN(t)}{100}) \qquad\qquad 2$$

The change of ATN over a time interval ($\Delta t$) for the instrument operated at a volume flow rate of
Q and spot area of A yields the attenuation coefficient ($B_{ATN}$) for that time interval:
$$B_{ATN} = \frac{A}{Q \times \Delta t} \times \Delta ATN \qquad\qquad 3$$

$B_{ATN}$ is finally converted to $B_{abs}$ by applying correction algorithms. The general form of the
correction algorithms presented for the PSAP in Bond et al. (1999) and Virkkula et al. (2005) can
be summarized as:
$$B_{abs} = B_{ATN} \times f(Tr) - C_1 \times B_{scat} \qquad\qquad 4$$

where f(Tr) is some function of Tr (that may vary between approaches), correcting for the filter
loading effect. $C_1$ is a constant that may vary with wavelength; specifically, it is a penalty for the
light scattering by the particles collected on the filter which may contribute to the quantification
of ATN. In most atmospheric and laboratory studies, $B_{scat}$ is measured independently, typically
using a co-located NEPH.
2.3.1. The B1999 correction
Bond et al. (1999) was the first study to present the correction algorithm for filter-based
instruments. This empirical correction was originally developed for the PSAP operated at 550 nm
using various mixtures of laboratory-generated nigrosin (SSA ≈ 0.5) and ammonium sulfate (SSA
≈ 1) with $B_{abs}$ ranged from 0 to 800 Mm$^{-1}$.
After calibrating the flow rate and spot area of the PSAP, the authors derived $C_1$ = 0.016 and
$$f(Tr)_{B1999} = \frac{1}{C_2 \times Tr + C_3} \qquad\qquad 5$$

where $C_2$ = 1.32 and $C_3$ = 0.87 (after combining Eq. (3) and Eq. (12) from Bond (1999)).
The equation parameters were further clarified in Ogren (2010) who adjusted the B1999-measured
spot area (A = 20.43 mm$^2$) to be consistent with the universal area of the PSAP (A = 17.83 mm$^2$).
Ogren (2010) also extended the correction to 574 nm using a wavelength dependence of $B_{abs}$ ($B_{abs}$
~ $\lambda^{-0.5}$). Consequently, $C_2$ and $C_3$ in f(Tr) were updated to 1.55 and 1.02, respectively. These are
the values used in the present work (Table 2) for B1999. Moreover, Ogren (2010) stated that the
correction forms of Eq. (4) and Eq. (5) were valid for any wavelength, while additional



experiments were needed to establish the equation parameters for the wavelengths other than 574
nm.

### 2.3.2. The V2005 correction

Virkkula et al. (2005) developed a correction algorithm for both three-wavelength PSAP (467, 530,
and 660 nm) and one-wavelength PSAP (574 nm) using the same functional form as Eq. (4). Since
the operating wavelengths of the photoacoustic instruments and the NEPH were different from
those of the PSAP, the measured photoacoustic $B_{abs}$ and $B_{scat}$ was extrapolated or interpolated to
467, 530, and 660 nm, using inferred AAE and SAE respectively. In this study, the authors used
various mixtures of kerosene soot, ammonium sulfate, and polystyrene latex (SSA ranged from
0.2 to 0.9) with $B_{abs}$ ranging from 0 to 800 Mm$^{-1}$ at 530 nm.
Different from the f(Tr) in the B1999 correction which was a reciprocal function of Tr, the f(Tr)
presented in V2005 was a multi-variate linear function of the natural logarithm of Tr and SSA
(including an interaction term between the two):

$$f(Tr(\lambda), SSA(\lambda))_{V2005} = C_4 + C_5 \times (C_6 + C_7 \times SSA(\lambda)) \times ln(Tr(\lambda)) \qquad 6$$

where the parameters in Eq. (6) vary with wavelengths. The parameters in V2005 were updated in
Virkkula (2010) by correcting for flowmeter calibration (Table 2).
Due to the unknown values of SSA before deriving $B_{abs}$, Virkkula et al. (2005) provided a solution
through an iterative procedure. In the iteration, $B_{abs}$ is first calculated using the B1999 correction
(e.g., Eq. (4) and Eq. (5)) and is then used to compute the initial guess of SSA for use in Eq. (6).
The $B_{abs}$ and SSA can be updated using Eq. (4) and Eq. (6) until convergence is reached.

### 2.3.3. The new correction

We develop a set of new correction algorithms with the same general form as Eq. (4) using the
biomass burning emissions from 65 different burns during the FIREX laboratory study, providing
a broader range of aerosol optical properties and aerosol concentrations than previous work. This
was motivated by the disagreement that remained between filter-based and photoacoustic
instruments, even after applying B1999 to the data (e.g., see Li et al. (2019) Fig. 4 and our Fig. 2
below). These differences may persist because we were effectively extrapolating the B1999
correction equation to values outside the range for which it was developed.
This new correction is developed based on multiple linear regression techniques with three
dependent variables of ln(Tr), SSA, and AAE and one independent variable of $B_{abs}/B_{ATN}$ (Eq. (7)
– (9)). As with other correction equations, this model takes into account the influence of scattering
and weakly-absorbing materials. However, we target two additional aims: 1) extend the correction
a wider range of $B_{abs}$; and 2) develop a model that is applicable to any filter-based instrument.
Similar to the B1999 and V2005 corrections, this new model starts with the general form of Eq.
(4), re-written here to define $B_{scat}$ in terms of SSA and $B_{abs}$.

$$B_{abs}(\lambda) = B_{ATN}(\lambda) \times f(Tr(\lambda)) - C_1 \times \frac{SSA(\lambda)}{1-SSA(\lambda)} \times B_{abs}(\lambda) \qquad 7$$

Re-arranging this equation to move all $B_{abs}$ terms to the left-hand side yields:

$$B_{abs}(\lambda) = B_{atn}(\lambda) \times g(Tr(\lambda), SSA(\lambda)) \qquad 8$$





where $g(Tr(\lambda), SSA(\lambda)) = f(Tr(\lambda)) \times \frac{1 - SSA(\lambda)}{1 - (1 - C_1) \times SSA(\lambda)}$.
We define the function "g" as a multivariate linear model, introducing AAE as a dependent
variable and including interaction terms between SSA, AAE, and ln(Tr):
$$g(Tr(\lambda), SSA(\lambda), AAE) = G_0 + G_1 \times ln(Tr(\lambda)) + G_2 \times SSA(\lambda) + G_3 \times AAE + G_4 \times ln(Tr(\lambda)) \times SSA(\lambda) +$$
$$G_5 \times ln(Tr) \times AAE + G_6 \times SSA(\lambda) \times AAE + G_7 \times SSA(\lambda) \times AAE \times ln(Tr(\lambda)) \quad 9$$
Equation (9) suggests that different combinations of SSA, AAE and ln(Tr) can result in the same
value of "g" (i.e., $B_{abs}/B_{ATN}$); likewise, a given value of $B_{abs}/B_{ATN}$ may have infinitely many points
with distinct slopes passing through it (Fig. S3). Therefore, in orderly to properly compensate for
the effects of loading and aerosol optical properties, a multiple linear regression with interaction
terms is required.
A detailed description of the procedure for the model development (e.g., variable transformation
(from Tr to ln(Tr)), variable selection using best-subsets and stepwise approaches, and model
validation) is provided in the Supplementary Material.
As in V2005, iteration is required in our algorithm because $B_{abs}$ is dependent on knowledge of
SSA and AAE, which themselves are dependent on $B_{abs}$. We propose the following iterative
process to update SSA and AAE in the model.
1. Initialize AAE from $B_{ATN}$ across the three wavelengths ($B_{ATN} \sim \lambda^{-AAE}$) and initialize SSA for
each wavelength using $B_{ATN}$ from the filter-based absorption photometer and $B_{scat}$ from a co-
located NEPH, i.e., $SSA(\lambda) = \frac{B_{scat}(\lambda)}{B_{scat}(\lambda) + B_{ATN}(\lambda)}$.
2. Yield an initial set of coefficients $G_0$ through $G_7$ for each wavelength to calculate g(Tr, SSA,
AAE) in Eq. (9), using one of the Algorithms described in Sect. 2.4.
3. Calculate $B_{abs}$ for each wavelength using Eq. (8).
4. Update AAE and SSA using $B_{abs}$ calculated in Step 3.
5. Derive a new set of coefficient values.
6. Iterate Steps 3-5 until converged.
2.4. Application of correction algorithms
In developing a procedure for applying our algorithm, we envision three potential scenarios:
1. Algorithm A: The filter-based instrument is co-located with a NEPH and reference instrument
providing $B_{abs}$. This scenario facilitates the computation of $G_0$ through $G_7$ in Eq. (9) (step 2 in
the iterative process) as well as the derivation of new coefficients for existing correction
algorithms. This scenario can also enable the develop of a new a set of coefficients that may
be more appropriate for aerosol sources that we do not consider here.
2. Algorithm B: The filter-based instrument is co-located with a NEPH but not a reference
instrument providing $B_{abs}$, which is perhaps the most likely scenario (at least at many long-
term monitoring sites). This scenario requires an initial guess of the coefficients; we provide
sets of these in Table 4 below for different filter-based instruments and aerosol sources.
3. Algorithm C: The filter-based instrument is deployed with neither a co-located NEPH nor a
reference instrument providing $B_{abs}$. This scenario is the most challenging, because there are
no measurements of $B_{scat}$ to compute SSA; to address this issue, we propose the use of a non-





linear relationship between SSA and AAE (AAE = a + b×SSA$^c$) to provide an initial guess of
SSA in the iterations.
To aid in decision-making between algorithms, we developed a flow chart for selecting appropriate
correction algorithm for CLAP, TAP, and PSAP (Fig. 1). Furthermore, an Igor Pro (WaveMetrics,
Inc.) based program for selecting and implementing our correction algorithms can be found in the
Supplemental Material.

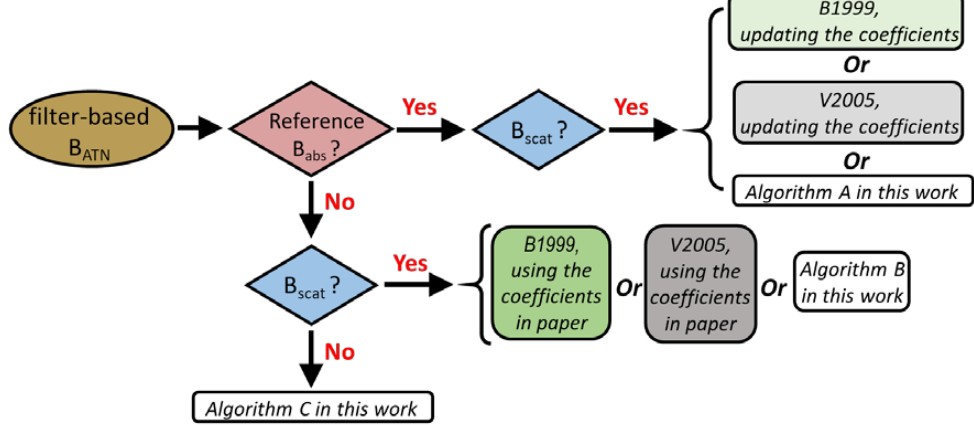


**Figure 1.** The flow-chart for the application of correction algorithms on PSAP, CLAP, and TAP.
Similar logic is followed for the AETH.
3.   Results and discussion
3.1. Application of the previous algorithms on different aerosols
We first consider the application of the B1999 and V2005 corrections on different combinations
of aerosol type and filter-based absorption photometer. Specifically, we apply the two corrections
to the biomass burning data from the FIREX laboratory campaign (CLAP and TAP) as well as six
months of ambient data from the SGP site (CLAP and PSAP). In doing so, we use the "default"
coefficients recommended in B1999 and V2005 as well as "updated" coefficients that are
estimated via regression techniques. We focus on the results of the CLAP in the main text, because
a CLAP is the only instrument common to deployments for both FIREX and SGP. The results of
the TAP from FIREX and the PSAP from the SGP site can be found in the Supplementary Material
(Table S5 and Fig. S5).
Our inter-comparison between the corrected CLAP-derived $B_{abs}$ and reference $B_{abs}$ for the FIREX
and SGP data is provided in Fig. 2 and Table 3. For the FIREX measurements, both analyses (using
the "default" coefficients and updating the coefficients) suggest good correlation (coefficient of
determination ($R^2$) > 0.9) between the CLAP and the reference across all three wavelengths.
Nevertheless, the corrections using the "default" coefficients result in over-prediction of $B_{abs}$ by
factors of ~2.5. If we update the coefficients in the corrections, there is an obvious improvement
in the agreement (i.e., slope ≈ 1; $R^2$ increases). The results are generally similar for SGP, although
the $R^2$ for ambient data is generally lower for ambient data ($R^2$ < 0.7). Decreased $R^2$ may be due
to the lower aerosol concentrations measured in ambient air, which could lead to lower signal-to-





noise in the instruments. Moreover, it is worth mentioning that for both datasets (FIREX and SGP),
the corrected $B_{abs}$ from different filter-based absorption photometers using the "default"
approaches does not agree with each other (slopes range from 0.69 to 1.40). However, after
updating the coefficients, the slopes approach unity (Table S6).

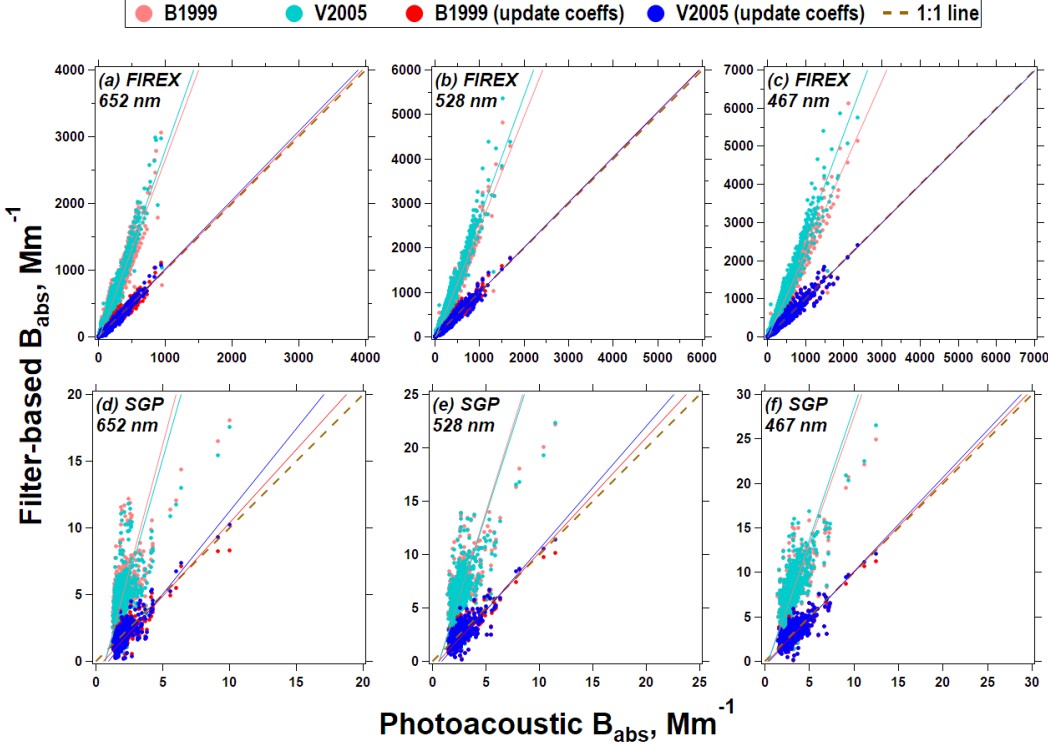


**Figure 2.** Inter-comparison between the CLAP-derived $B_{abs}$ corrected by the B1999 and V2005
algorithms and the reference $B_{abs}$ at 652, 528, and 467 nm for both FIREX and SGP data. The solid
lines represent linear regressions, while the dashed line is a 1:1 line.
**Table 3** Relationship between the CLAP-derived $B_{abs}$ corrected by the B1999 and V2005
algorithms (including updated coefficients) and the reference $B_{abs}$ at 652, 528, and 467 nm. The
relationship is achieved using major axis regression (Ayers, 2001). The value in parentheses
represents the coefficient of determination ($R^2$) of the linear relationship.

|  |  | 652 nm | 528 nm | 467 nm |
|---|---|---|---|---|
| FIREX | B1999 | y = -39 + 2.69x (0.94) | y = -49 + 2.50x (0.96) | y = -45 + 2.26x (0.97) |
|  | V2005 | y = -46 + 2.83x (0.96) | y = -57 + 2.75x (0.96) | y = -56 + 2.68x (0.96) |
|  | B1999 (update coeffs) | y = -8.4 + 1.02x (0.96) | y = -7.7 + 1.01x (0.97) | y = -3.4 + 1.00x (0.96) |
|  | V2005 (update coeffs) | y = -9.4 + 1.03x (0.97) | y = -7.3 + 1.01x (0.97) | y = -3.0 + 1.00x (0.96) |
| SGP | B1999 | y = -2.60 + 3.77x (0.41) | y = -1.90 + 3.20x (0.49) | y = -0.98 + 2.85x (0.55) |
|  | V2005 | y = -2.50 + 3.54x (0.41) | y = -2.00 + 3.15x (0.48) | y = -1.10 + 2.96x (0.55) |
|  | B1999 (update coeffs) | y = -0.29 + 1.10x (0.60) | y = -0.29 + 1.08x (0.63) | y = -0.17 + 1.03x (0.65) |
|  | V2005 (update coeffs) | y = -0.57 + 1.24x (0.65) | y = -0.50 + 1.15x (0.67) | y = -0.27 + 1.06x (0.67) |





In the FIREX data, there is an apparent dependency of the updated coefficients on the wavelength
of light, but more importantly, on the aerosol optical properties, namely SSA and AAE (Tables
S7-S9). However, in the ambient data from SGP, the dependency on optical properties is less
obvious (Tables S10-S11). Nevertheless, all of these coefficients differ from those reported in
B1999 and V2005 (again, derived for the PSAP rather than the CLAP), which highlights the
potential need to use coefficient values that are appropriate for the instrument being used, its
wavelength(s) of light, and optical properties that are representative of the sampled aerosols when
applying correction factors to $B_{ATN}$.
3.2. Application of the new algorithms to the FIREX data
The co-location of the CLAP, TAP, AETH, and PAX during FIREX allows us to apply each
algorithm (A, B, C) to these data. Similar to Sect. 3.1, we focus our discussion on the CLAP with
details on the TAP and AETH presented in the Supplementary Material (Fig. S5-S6). However,
we provide the recommended initial guesses in the new algorithms and the comparison of
absorption (corrected filter-based $B_{abs}$ versus reference $B_{abs}$) for all filter-based absorption
photometers in Table 4 and Table 5 to help readers quickly retrieve key information of our
algorithms.
Figure 3 provides a comparison between the uncorrected $B_{ATN}$ from the CLAP at all three
wavelengths, as well as photoacoustic $B_{abs}$ interpolated to those wavelengths using AAE. For each
wavelength, the slopes are significantly greater than one. Moreover, there is an apparent
dependency on SSA and AAE in the agreement between the instruments. This is most obvious in
Fig. 3a (652 nm), where data with lower SSA and lower AAE (smaller markers, "brighter" colors)
fall below the best-fit line, while data with higher SSA and higher AAE (larger markers, "darker"
colors) fall above the best-fit line. This phenomenon is less clear in Fig. 3b-3c, but an apparent
dependancy on SSA and AAE remains, which highlights the need to include both of these aerosol
optical properties (and appropriate interaction terms) when correcting $B_{ATN}$ values.

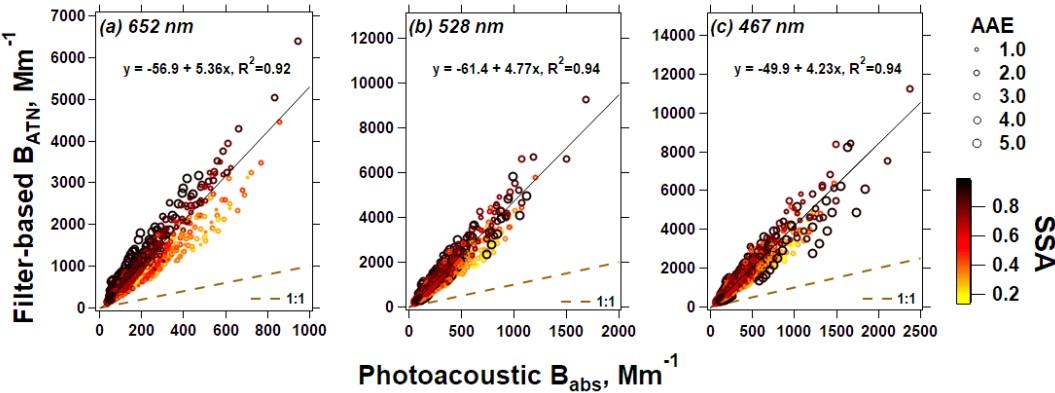


**Figure 3.** Comparison of the uncorrected CLAP-derived $B_{ATN}$ and the reference $B_{abs}$ at 652, 528,
and 467 nm for the FIREX data. The data points are colored by the corresponding SSA. The size
of data points reflects their AAE quantified by the two PAX. The solid line represents the linear
regression, while the dashed line is a 1:1 line.





We first apply "Algorithm A" to the CLAP $B_{ATN}$ data in Fig. 3. Using the reference $B_{abs}$ values
from the PAX (in addition to $B_{scat}$ values), we are able to derive a set of coefficients that enable
the correction of the data (Table 4). Corrected CLAP values are presented in Fig. 4 with the linear
relationships presented in Table 5. The slope for each wavelength is very close to the 1:1 line,
suggesting that our approach works well in correcting these data. Moreover, the heteroscedasticity
that exists in Fig. 3 has been minimized after correction, and there are no apparent trends in how
the data are organized in Fig. 4 due to the aerosol optical properties.
**Table 4** Coefficient values for Eq. (9) derived using "Algorithm A". We recommend these as the
initial guesses when implementing "Algorithm B".

| | | $G_0$ | $G_1$ | $G_2$ | $G_3$ | $G_4$ | $G_5$ | $G_6$ | $G_7$ |
|---|---|---|---|---|---|---|---|---|---|
| CLAP (FIREX) | 652 nm | 0.27 | -0.16 | -0.18 | -0.05 | 0.18 | 0.08 | -0.01 | 0.03 |
| | 528 nm | 0.30 | -0.28 | -0.18 | -0.07 | 0.25 | 0.10 | 0.13 | -0.17 |
| | 467 nm | 0.32 | -0.38 | -0.20 | -0.08 | 0.33 | 0.12 | 0.24 | -0.31 |
| TAP (FIREX) | 652 nm | 0.45 | -0.45 | 0.07 | -0.19 | 0.94 | 0.10 | 0.26 | -0.35 |
| | 528 nm | 0.54 | -0.51 | 0.02 | -0.26 | 0.76 | 0.20 | 0.38 | -0.44 |
| | 467 nm | 0.62 | -0.59 | -0.07 | -0.32 | 0.73 | 0.29 | 0.53 | -0.60 |
| CLAP (SGP) | 652 nm | 0.37 | -0.18 | -0.34 | -0.11 | 0.30 | 0.18 | -0.36 | 0.41 |
| | 528 nm | 0.40 | -0.15 | -0.42 | -0.14 | 0.10 | 0.24 | -0.17 | 0.25 |
| | 467 nm | 0.43 | -0.16 | -0.45 | -0.16 | 0.07 | 0.27 | -0.06 | 0.12 |
| PSAP (SGP) | 652 nm | 0.24 | 0.35 | -0.16 | -0.04 | -0.47 | 0.07 | -0.57 | 0.73 |
| | 528 nm | 0.30 | 0.48 | -0.26 | -0.10 | -0.67 | 0.17 | -0.63 | 0.77 |
| | 467 nm | 0.35 | 0.49 | -0.34 | -0.15 | -0.69 | 0.23 | -0.55 | 0.79 |
| AETH (FIREX) | 950 nm | 0.47 | 0.17 | 0.01 | -0.27 | -0.4 | 0.25 | -0.12 | 0.27 |
| | 880 nm | 0.34 | 0.13 | 0.13 | -0.17 | 0 | 0.10 | -0.13 | 0.12 |
| | 660 nm | 0.28 | 0.09 | 0.11 | -0.12 | 0.15 | 0.05 | -0.12 | 0.03 |
| | 590 nm | 0.16 | -0.08 | 0.26 | -0.03 | 0.59 | -0.08 | -0.02 | -0.19 |
| | 520 nm | 0.16 | -0.05 | 0.14 | -0.01 | 0.54 | -0.07 | -0.02 | -0.21 |
| | 470 nm | 0.14 | -0.05 | 0.06 | 0 | 0.53 | -0.05 | -0.02 | -0.17 |
| | 370 nm | 0.13 | -0.09 | 0.11 | 0 | 0.59 | -0.06 | -0.01 | 0.01 |


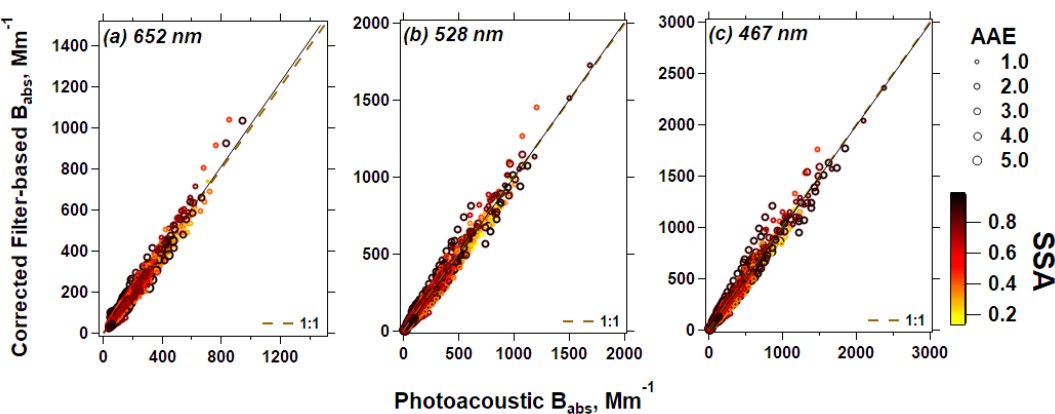


**Figure 4.** As in Fig. 3, but the CLAP-based $B_{ATN}$ values have been corrected using our "Algorithm
A".





**Table 5** Relationship between the filter-based $B_{abs}$ corrected by "Algorithm A" and the reference
$B_{abs}$ at the operating wavelengths for the filter-based instrument. The relationship is achieved using
major axis regression (Ayers, 2001). The value in the parentheses represents the coefficient of
determination ($R^2$) for the linear relationship.

| | | 652 nm | | 528 nm | | 467 nm | |
|---|---|---|---|---|---|---|---|
| FIREX | CLAP | y = -7.8 + 1.02x (0.98) | | y = -6.2 + 1.01x (0.98) | | y = -3.2 + 1.00x (0.98) | |
| | TAP | y = -10 + 1.00x (0.87) | | y = -13 + 0.99x (0.87) | | y = -16 + 0.99x (0.88) | |
| SGP | CLAP | y = -0.25 + 1.08x (0.68) | | y = -0.21 + 1.05x (0.67) | | y = -0.04 + 0.99x (0.68) | |
| | PSAP | y = -0.28 + 1.10x (0.43) | | y = -0.24 + 1.06x (0.55) | | y = -0.07 + 1.00x (0.62) | |
| FIREX | AETH | 950 nm | 880 nm | 660 nm | | 590 nm | |
| | | y = -3.19 + 1.01x (0.82) | y = -3.92 + 1.02x (0.85) | y = -5.97 + 1.03x (0.88) | | y = -5.63 + 1.02x (0.90) | |
| | | 520 nm | 470 nm | 370 nm | | - | |
| | | y = -2.36 + 0.99x (0.90) | y = 2.93 + 0.95x (0.88) | y = 18.38 + 0.89x (0.80) | | - | |


We next investigate the repeatability of the coefficient values presented in Table 4 by randomly
selecting half of the measurements (N = 1338) from the whole FIREX dataset. By implementing
"Algorithm A" to the extracted observations, we obtain new coefficient values for $G_0$ to $G_7$. This
is repeated 1000 times to obtain a distribution of coefficient values (Fig. S7). The extraction
approach mimics the process of obtaining new biomass burning datasets, so that we can estimate
the variability of these derived coefficients. From Fig. S7 and Table S12, the derived coefficients
are mostly insensitive to the different randomly-extracted datasets; most of the quartile deviation
(defined as (Q3-Q1)/2, where Q1 and Q3 are the first and third quartile respectively) is within 0.05,
except $G_4$ which has a quartile deviation of ~0.08. Consequently, the coefficient values obtained
in "Algorithm A" appear to be reasonable initial guesses to correct filter-based absorption
measurements during biomass burning events when the reference $B_{abs}$ is unavailable, such as in
"Algorithm B" and "Algorithm C".
We next implement "Algorithm B" to the CLAP $B_{ATN}$ data from Fig. 3 using the initial guesses of
the coefficients derived from "Algorithm A" (Table 4) along with reference $B_{scat}$ values. To get a
sense of the variability in the results, we randomly select half of the data and applied the correction;
this process is repeated 1000 times. For each iteration, we compare the corrected $B_{abs}$ from the
CLAP to the reference $B_{abs}$ from the PAX; the resulting slope, intercept, and $R^2$ values are
summarized as box-and-whisker plots in Fig. 5. For all three wavelengths, the slopes are close to
unity, and there is good correlation between the two absorption measurements ($R^2 \approx 0.98$), which
indicates that the good performance seen in Fig. 4 is independent of the reference $B_{abs}$
measurements and our algorithm is able to correct "new" $B_{ATN}$. Consequently, when scattering
measurements are co-located with filter-based absorption measurements, our new correction
algorithm performs well.

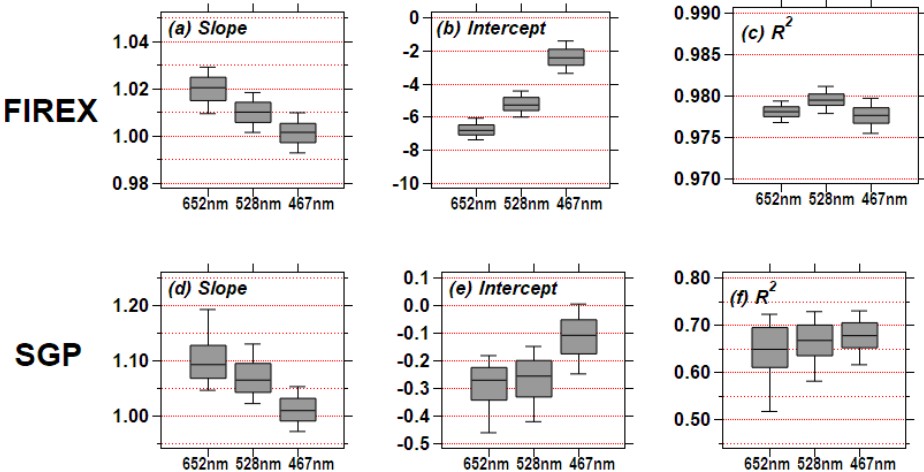

**Figure 5.** The box-and-whisker plots for the slope, intercept, and $R^2$ of the relationship between the CLAP-derived $B_{abs}$ (corrected by "Algorithm B" in the present work) and PAX-derived $B_{abs}$ for all three wavelengths. For details on how these values were generated, please refer to the text.

Lastly, we apply "Algorithm C" to the data in Fig. 3. However, we first require a functional relationship between AAE and SSA, because in this scenario, the CLAP $B_{ATN}$ values are the only data input to the algorithm (and therefore, SSA is unknown). Liu et al. (2014) proposed that a power function can describe this relationship (AAE = $a + b \times SSA^c$); we present these data from FIREX along with power function fits (and associated prediction intervals) in Fig. 6. To define AAE in this figure, we fit a power-law relationship to the three $B_{ATN}$ values from the CLAP; similarly, we define SSA using interpolated $B_{scat}$ from the PAX and $B_{ATN}$ from the CLAP (The rationale for using $B_{ATN}$ is that if "Algorithm C" were to be implemented in practice, only $B_{ATN}$ would be available). In Fig. 6, the data points are colored by "prediction error", effectively a metric to quantify how well the power function reproduces the individual data points. Although there is a fair amount of error in some of these points, we still obtain an SSA-AAE relationship required to initialize "Algorithm C".

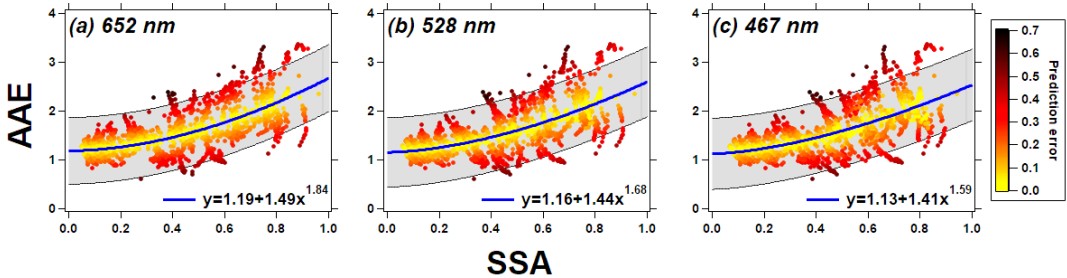

**Figure 6.** AAE plotted against SSA for the FIREX data. In the figures, AAE was computed using a power-law fit across all three wavelengths, and SSA was computed using the interpolated $B_{scat}$ from the two PAX and the reported $B_{ATN}$ from the CLAP. The data points are colored by their prediction error (("true" AAE - "calculated" AAE)/ "calculated" AAE).



Even though there is uncertainty in the SSA vs. AAE relationship used in "Algorithm C", after
corrections have been applied, the filter-based $B_{abs}$ for the CLAP agrees well with the independent
reference $B_{abs}$; the slopes for all wavelengths are slightly greater than 1 (1.03-1.05) and the $R^2$
values are all high (0.97-0.98). However, even though the absorption measurements are corrected
well, there still remains large uncertainties in values of inferred scattering. Examples of this are
provided in Fig. 7, where we compare the SSA inferred from the PAX to the SSA inferred from
"Algorithm C" as well as $B_{scat}$ for each wavelength. Generally speaking, data that are better
represented by the SSA vs. AAE relationship (i.e., smaller prediction error) results in better
agreement with the reference for both SSA and $B_{scat}$, but there is also a clear divergence from the
1:1 line in Fig. 7a-c as SSA decreases. Therefore, even though "Algorithm C" performs well at
correcting filter-based $B_{ATN}$ to agree with the reference $B_{abs}$, estimates of final SSA values should
be considered to be uncertain.

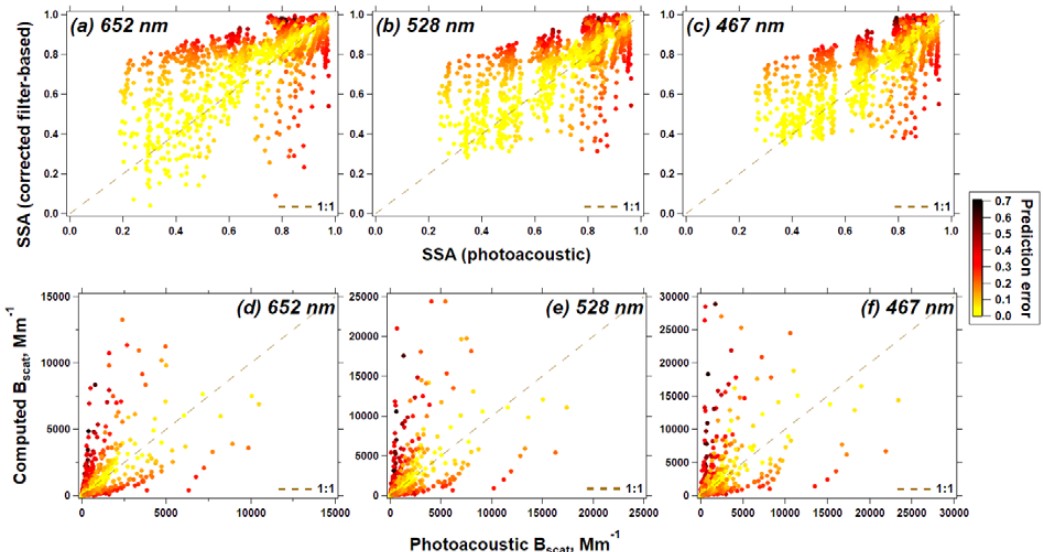


**Figure 7.** Comparison of SSA (a-c) and $B_{scat}$ (d-f) at the three wavelengths for the FIREX data.
Vertical axis: values output from "Algorithm C"; horizontal axis: values calculated using the
photoacoustic $B_{abs}$ and $B_{scat}$.
In addition to the CLAP, we apply the new algorithms to the other filter-based absorption
photometers operated during the FIREX study (TAP and AETH). Consistent with what we
observed for the CLAP results, the corrected TAP- and AETH-derived $B_{abs}$ is in good agreement
with the photoacoustic $B_{abs}$ (as demonstrated in Table 4 and Table 5, as well as Fig. S5-S6).
Moreover, the corrected $B_{abs}$ from the three filter-based instruments agrees with each other for all
three wavelengths (Table 6), confirming the universal nature of our algorithm.





**Table 6** Inter-comparison between different filter-based $B_{abs}$ corrected by "Algorithm A" in the
present work. The value in the parentheses represents the coefficient of determination ($R^2$) of the
linear relationship.

| | FIREX: CLAP vs. TAP | FIREX: CLAP vs. AETH | FIREX: TAP vs. AETH | SGP: CLAP vs. PSAP |
|---|---|---|---|---|
| 652 nm | y = 1.84 + 1.02x (0.89) | y = 4.17 + 0.94x (0.87) | y = -0.31 + 0.99x (0.82) | y = -0.04 + 0.99x (0.70) |
| 528 nm | y = 5.75 + 1.02x (0.88) | y = 3.70 + 0.91x (0.85) | y = -6.38 + 0.98x (0.82) | y = -0.11 + 1.02x (0.73) |
| 467 nm | y = 10.57 + 1.01x (0.88) | y = 0.45 + 0.98x (0.83) | y = -13.62 + 1.04x (0.79) | y = -0.11 + 1.02x (0.76) |


3.3. Application of the new algorithms to ambient data
To test our algorithms further, we extended our work to ambient data collected the DOE SGP site
during the time period which the PASS-3 was operational. From the SGP data, we derived a
different set of coefficients for ambient data using "Algorithm A", which differ from those derived
for FIREX (Table 4). The results presented in Fig. 8 and Table 5 suggest that our new algorithm
works at least as well as B1999 and V2005 on this dataset (both with updated coefficients). The
repeatability of the coefficient values in "Algorithm A" is confirmed for the SGP measurements
using the same procedure as described in Sect. 3.2 (see results in Fig. S7 and Table S12).

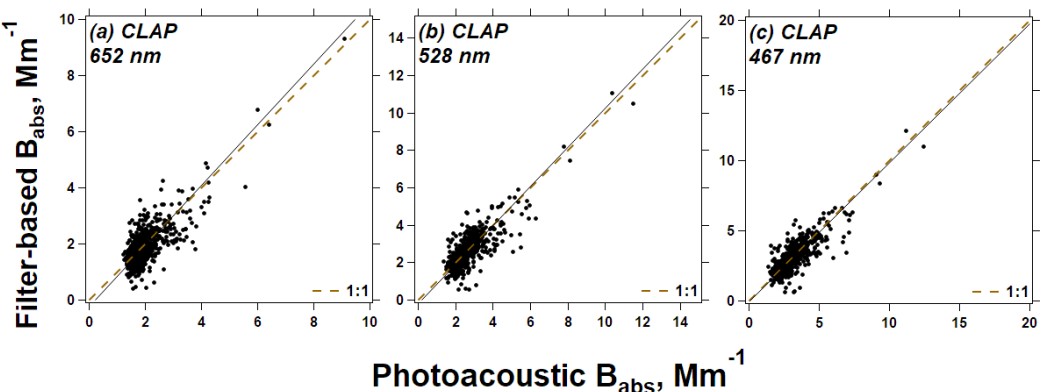


**Figure 8.** Inter-comparison between the CLAP-derived $B_{abs}$ corrected by "Algorithm A" in the
present work and reference $B_{abs}$ at 652, 528, and 467 nm for the ambient data at the SGP study
area. The solid line represents the linear regression, while the dashed line is a 1:1 line.
On the SGP data, we see similar performance to the FIREX data when we apply "Algorithm B",
where we again sampled half of the CLAP data, used the initial guesses derived in "Algorithm A",
and repeated this process 1000 times. Although the slopes tend to be larger than 1 (i.e., the
corrected CLAP $B_{abs}$ remains high relative to the PASS $B_{abs}$), the results still represent an
improvement over B1999 and V2005 using their recommended coefficients for their correction
equations.
Implementing "Algorithm C" is challenging for ambient data, because there is no distinct power
function relationship in AAE vs. SSA (Fig. 9); this is consistent with other field studies reporting
both SSA and AAE (e.g., Backman et al. (2014) and Lim et al. (2018)). Our approach described
here is only appropriate for ambient aerosols that follow a power function, such as sites impacted





by biomass burning. Nevertheless, we did apply this to a subset of the SGP data where the AAE-
SSA prediction error is within 30% (N = 86), and for this subset of data, "Algorithm C" works
fairly well (slopes ≈ 0.95; see Fig. S8). Therefore, while "Algorithm C" may have utility for
ambient data, we advise caution when using this algorithm since the aerosols influencing the site
may not be represented by a clear AAE-SSA power function (e.g., when biomass burning and
coarse aerosols are equally prevalent at a long-term monitoring site).

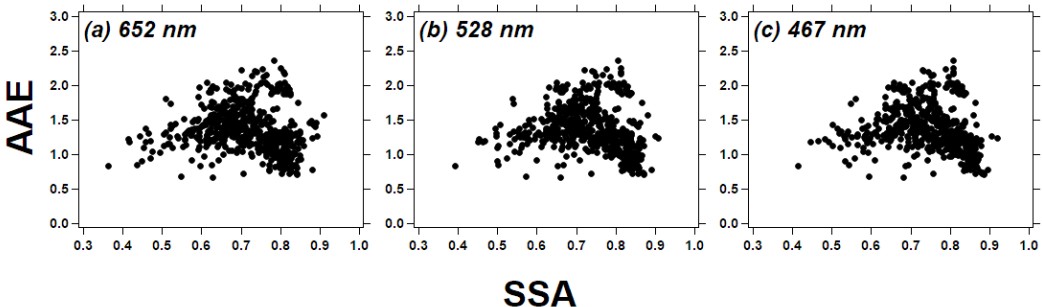


**Figure 9.** AAE plotted against SSA for the SGP ambient data. The power law fit (AAE = a + b × SSA$^c$)
is performed on SSA (SSA = $B_{scat}/(B_{scat} + B_{ATN})$) and AAE computed by three-wavelength $B_{ATN}$.

These new algorithms are also applicable to the PSAP deployed at the SGP site. The results of the
correction for the PSAP are presented in Table 5 and Fig. S5, and the recommended initial guesses
when implementing "Algorithm B" to PSAP-$B_{ATN}$ at ambient environments are given in Table 4.
As expected, there is good agreement between corrected PSAP- and CLAP-$B_{abs}$ (Table 6).
3.4. Impact of the implemented correction algorithm on aerosol optical properties
In addition to the direct comparisons of $B_{abs}$ between the filter-based and photoacoustic
measurements, we compare derived optical properties (AAE and SSA) from different instruments
to assess the algorithms' performance on derived aerosol optical properties. For example, we have
discussed the discrepancy of SSA between the filter-based and photoacoustic measurements when
implementing "Algorithm C" in Sect. 3.2. In this section we will more broadly discuss the impact
of different correction algorithms on AAE and SSA.
In Fig. 10, we present the frequency distribution of AAE for both FIREX and SGP data generated
from different campaign/instrument pairs using different correction approaches. For the FIREX
data (Fig. 10a-b), most corrections (with the exception of the "default" B1999) are consistent with
the photoacoustic data, while for the SGP data (Fig. 10c-d), most corrections (with the exception
of "default" V2005) are consistent with the photoacoustic data. However, updating the coefficients
for B1999 and V2005 improves the agreement with the photoacoustic data. The 50% difference
that exists between the B1999 and V2005 algorithms in all panels in Fig. 10 are consistent with
previous studies. For example, both Backman et al. (2014) and Davies et al. (2019) found that the
V2005-derived AAE is greater than B1999-derived AAE by 33% to 50% for ambient aerosols.
Therefore, we highlight that the default coefficients in B1999 and V2005 may have some
limitations when deriving AAE using the corrected $B_{abs}$; instead, updating the coefficients or using
the new algorithm proposed in this work may yield more robust AAE results.

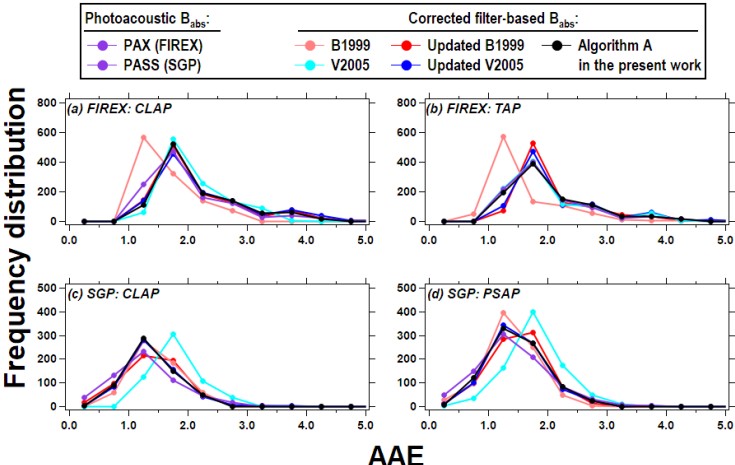


**Figure 10.** The frequency distribution of AAE calculated for different instrument/correction combinations of multi-wavelength $B_{abs}$.

Similar to Fig. 10, we also investigate the distribution of SSA computed by using corrected $B_{abs}$ along with $B_{scat}$. We provide the results at 652 nm as an example in the main text (Fig. 11); figures for 528 nm and 467 nm can be found in the Supplementary Material (Fig.S9 and S10). For both FIREX and SGP data, the SSA obtained using the new algorithm agree very well with the B1999 and V2005 but only when their coefficients have been updated. Calculations of SSA using B1999 and V2005 with their recommended coefficients suggest that these values may be biased low, which follows the over-estimation of corrected $B_{abs}$ demonstrated in Fig. 2.

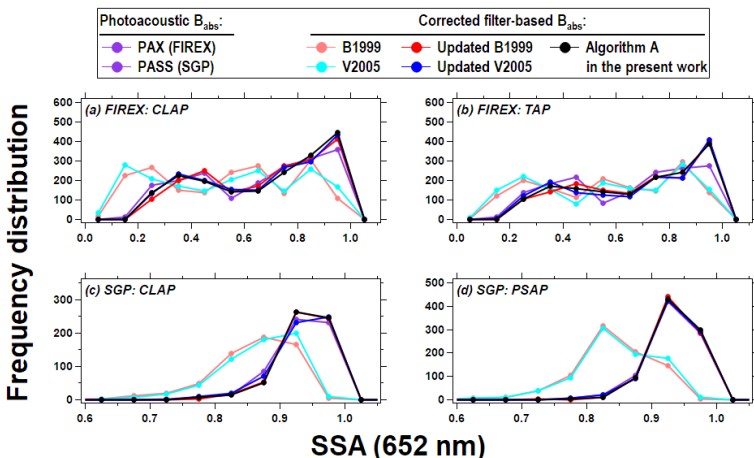

576

**Figure 11.** The frequency distribution of SSA (652 nm) calculated for different instrument/correction combinations of $B_{abs}$ and $B_{scat}$.

Moreover, we plot similar figures as Fig. 10-11 using all algorithms (A, B, and C). As shown in Fig. S11, the results using "Algorithm B" agrees very well with those using "Algorithm A", but


the use of "Algorithm C" results in some obvious discrepancies compared to the photoacoustic
reference, again highlighting the potential for large uncertainty using this algorithm.
In Fig. 12, we directly compare the distributions of both AAE and SSA at 652 nm for all of the
filter-based absorption photometers considered here, using our "Algorithm A" to correct the $B_{ATN}$
data. For both datasets, after the corrections have been applied, there are only marginal differences
of the AAE (Fig. 10a and 10b) derived by different instruments. Similarly, there is good agreement
among the SSA values when using corrected-$B_{abs}$ from different instruments (Fig. 10c and 10d).
Overall, the derived properties using the new correction are consistent across all instruments,
suggesting its universality.

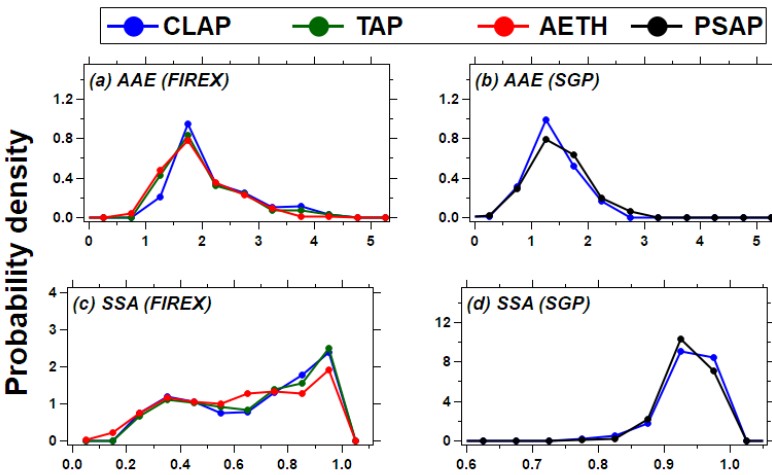


**Figure 12** The probability density of AAE and SSA (652 nm) derived by different filter-based
photometers $B_{abs}$ (corrected by "Algorithm A" in the present work). Note that the number of total
observations vary across instruments.
3.5. Uncertainty of the new algorithms
In this section, we estimate the uncertainty of the new algorithms due to both measurement
uncertainties of the instruments and the uncertainties of parameter computation. We then simulate
the propagated uncertainty in the corrected filter-based $B_{abs}$ reported in this paper.
Measurement uncertainties of the instruments considered here have been reported in previous work
(e.g., (Anderson et al., 1996; Nakayama et al., 2015; Ogren et al., 2017; Sherman et al., 2015)) and
are summarized in Table 1. The typical sources of measurement uncertainty of the aerosol
instruments include: 1) instrument noise (often associated with the averaging time); 2) calibration
uncertainties (such as the accuracy of the operating wavelengths and the properties of the
calibration materials); 3) standard temperature and pressure (STP) correction uncertainties
(Sherman et al., 2015); and 4) flow rate uncertainties. Additional uncertainties that are specific to
filter-based absorption photometers include spot size and filter medium corrections (Bond et al.,
1999; Ogren et al., 2017). Regardless, these values all tend to be ≤ 30%, which is consistent with
other commonly-used aerosol instrumentation.



Because correction algorithms for filter-based absorption instruments also require aerosol optical
properties, the algorithms' performance will be affected by these values as well. For example,
uncertainties in SSA are directly related to uncertainties associated with $B_{abs}$ and $B_{scat}$, which are
both included in our simulations. However, capturing uncertainties in AAE is more complex, as
AAE can be computed by either "2λ fit" (a linear fit using $B_{abs}$ at two wavelengths) or "3λ fit"
(same as the power fit used in the present work). Davies et al. (2019) used the 3λ fit to calculate
AAE and compared this to calculations using 662 nm and 785 nm (i.e., $AAE_{662/785}$), finding that
the 3λ results was about 50% greater. Moreover, similar differences (-35% to 85%) can exist
comparing two different 2λ combinations ($AAE_{440/870}$ and $AAE_{675/870}$), depending on the
contribution of brown carbon to absorption at 440 nm (Wang et al., 2016). However, based on Fig.
S12 and S13, we demonstrate small ( < ~10%) differences in the calculated values of AAE using
our Algorithm A using different 2λ combinations for linear fits and the 3λ power-law fit, when
considering both FIREX and SGP data. Consequently, we do not include AAE calculation
uncertainty in our simulation.
In our simulations, the propagated uncertainty of corrected $B_{abs}$ is estimated by implementing the
new algorithm to datasets in which filter-based $B_{ATN}$, reference $B_{abs}$, and $B_{scat}$ are subject to
measurement uncertainties. The full procedure is outlined in the Supplementary Material, but we
provide a brief overview of our Monte Carlo approach here. First, we create a synthetic dataset ($n$
= 500 records) that defines $B_{abs}$ at 652 nm and AAE that is intended to represent biomass burning.
Values of $B_{ATN}$ and SSA are then computed using the relationships presented in Fig. 3 and Fig. 6,
respectively. Respective uncertainties associated with each of these values are applied following
Table 1, assuming that these follow a normal distribution. We then applied "Algorithm B" to the
$B_{ATN}$ dataset, repeated 1000 times, to quantify overall uncertainty associated with our correction
algorithm.
Figure 13 provides a graphical summary of our uncertainty simulation results, which was derived
by fitting linear equations to the "true" $B_{abs}$ value (that we defined) and the "corrected" $B_{abs}$ values
(outputs of each iteration). Considering the slopes (Fig. 12a), our algorithm can generally
reproduce the "true" value within 10% at 652 nm and 528 nm, but the performance is slightly
degraded at 467 nm. The median intercept for our simulations is close to zero, but the interquartile
range increases with decreasing wavelength (Fig. 12b), suggesting that the uncertainty may
increase at shorter wavelengths. The coefficients of determination (Fig. 12c) range from 0.47 (652
nm) to 0.68 (467 nm), showing that the algorithm may be less precise if large measurement
uncertainties exist. Even though these sources of uncertainty exist when implementing our
correction algorithms and propagate through to the corrected values, we argue that our new
algorithm will "standardize" uncertainties across corrected $B_{abs}$ values from filter-based absorption
photometers. Moreover, the new algorithms perform, at least, better than the previous algorithms
with "default" coefficients, or as well as the previous algorithms with updated coefficients.

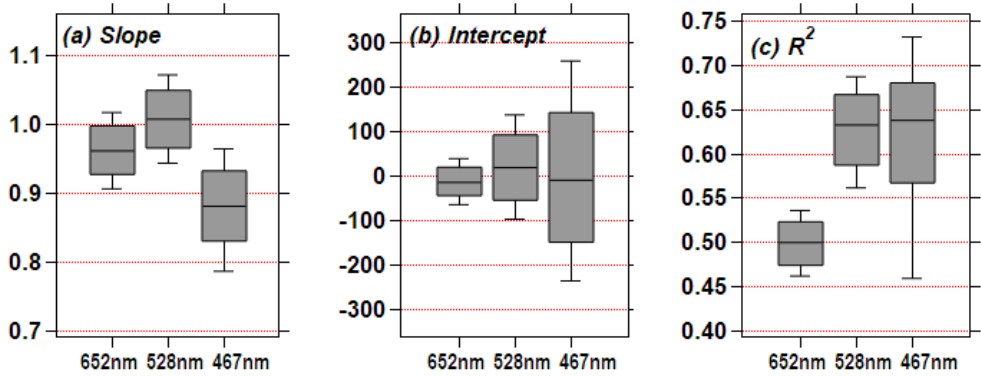


**Figure 13.** The box-and-whisker plots (slope, intercept, and $R^2$) for the Monte Carlo simulation of
the relationship between the CLAP-derived $B_{abs}$ (corrected by "Algorithm B" in the present work)
and "true" $B_{abs}$ for all three wavelengths.

4. Conclusions

Filter-based absorption instruments are widely used at global observational sites due to their
relatively low cost, fast response, and easy operation. Despite the existence of different correction
algorithms to correct the filter-based $B_{abs}$ measurements, these are not "standardized" as
differences in corrected $B_{abs}$ values exist across different instrument/correction combinations, even
when the instruments are co-located. This study provides a systematic evaluation of the previous
correction algorithms (B1999 and V2005 corrections) on the CLAP and similar instruments (TAP
and PSAP) using both laboratory-generated biomass burning emissions and ambient aerosols. We
also developed "universal" correction algorithms that are applicable to any filter-based absorption
photometer (e.g., PSAP, CLAP, TAP, AETH), which will have utility for any historic or future
filter-based absorption measurements and which have the potential to standardize absorption
coefficients across all filter-based instruments. This latter point is demonstrated in Table 6 and Fig.
12 in that there is good agreement across all filter-based absorption photometers when applying
our corrections to both biomass burning and ambient data. In practice, we anticipate that our
Algorithm B will be most common, because at long-term monitoring sites, filter-based absorption
photometers are typically co-located with a nephelometer.

Using the existing corrections on our CLAP measurements, we find that the corrected $B_{abs}$
overestimate photoacoustic $B_{abs}$ by factors of ~2.6 (biomass burning aerosols) and ~3.2 (ambient
aerosols). Similar overestimations of absorption by filter-based instruments are seen in the results
of TAP from the FIREX study and PSAP deployed at the SGP. Comparing between B1999 and
V2005, $B_{abs}$ corrected by the two corrections differ by -6% to 18%. These discrepancies in our
results are consistent with those reported for the inter-comparisons between filter-based and
photoacoustic absorption instruments (e.g., (Arnott et al., 2003; Davies et al., 2019; Li et al., 2019;
Müller et al., 2011a)).

Overall, our new developed algorithms (A, B, and C) perform well on correcting $B_{abs}$ for different
filter-based absorption photometers (CLAP, TAP, PSAP, and AETH) from both biomass burning


and ambient measurements. Our work suggests that if the filter-based instrument is co-operated
with a reference absorption instrument and a NEPH at field for a period, researchers can compute
site-specific initial guesses (same as "Algorithm A" in the present work). Otherwise, either
"Algorithm B" or "Algorithm C" proposed in this paper can be used to correct the filter-based
measurements. In "Algorithm B" when a filter-based absorption photometer is co-located with a
NEPH but without a reference instrument, the set of coefficients yield in this work (Table 4) can
be used as initial guesses to implement the algorithm. In "Algorithm C" when a filter-based
absorption photometer is operated by itself, a "representative" relationship between AAE and SSA
can be used to estimate SSA from AAE at each step in the iterative process, but we advise caution
if this relationship is not monotonic (e.g., as in the ambient data from SGP and from Backman et
al. (2014) and Lim et al. (2018)). The only scenario not included in the present work is that the
filter-based absorption photometer is co-located with a reference absorption instrument, but no
instrument for scattering. However, under this scenario, one could simply use the photoacoustic
$B_{abs}$ data because no filter-induced biases exist for those instruments.
In terms of the aerosol optical properties (AAE and SSA) computed by different corrections, the
new algorithm suggests no bias of AAE and SSA when compared to that derived by updated-
B1999and updated-V2005 for both aerosol datasets.
However, the new algorithm is not without limitations. First, we used the photoacoustic $B_{abs}$ as the
reference to develop the algorithm and the initial guess of the coefficients; meanwhile, some
studies argue that photoacoustic absorption is not a "ground truth" (e.g., (Lack et al., 2006; Lewis
et al., 2008)). Thus, we simulate the propagated uncertainty of our algorithms considering the
measurement uncertainties due to the photoacoustic $B_{abs}$ (as well as $B_{ATN}$ and $B_{scat}$) and find that
the corrected $B_{abs}$ can be biased by -17% to 5%, depending on the operated wavelength. Although
potential bias due to the precision of photoacoustic $B_{abs}$ cannot be excluded, using the universal
algorithm to correct the filter-based $B_{abs}$ will at least eliminate correction-related biases among
different filter-based instruments. Second, we only tested the algorithms with data from biomass
burning and ambient measurements. It is unclear how the algorithms will work for other absorbing
aerosols (e.g., dominated by fossil fuel emissions or mineral dust). Further evaluation of the
performance of the new algorithm on other aerosol sources may help to address this issue.
Regardless, we argue that our approach can standardize reported absorption coefficients at long-
term monitoring sites, which has the potential to yield a better data set with which to evaluate
chemistry-climate models.
***Code and data availability.*** The code for the algorithm has been developed in Igor Pro
(WaveMetrics Inc.). The package is available and fully described in the Supplementary Material.
The FIREX aerosol products are available at https://esrl.noaa.gov/csd/project/firex. The SGP
aerosol products are available at https://www.archive.arm.gov/discovery/ (February 2013–July
2013, 36° 36′ 18.0″ N, 97° 29′ 6.0″ W: Southern Great Plains Central Facility, data set accessed
712    01/16/19).

***Author contribution.*** AAM and GRM designed experiments, and HL and GRM conducted
experiments. HL conducted data analysis and developed the correction algorithm presented herein.
AAM and HL discussed the results. HL led the efforts to write the manuscript. All authors
contributed to the final manuscript.
***Competing interests.*** The authors declare that they have no conflict of interest.



***Acknowledgements.*** The project was funded by NOAA Climate Program Office Grant
NA16OAR4310109. The ambient data at SGP site in Lamont (OK, USA) were obtained from the
atmospheric radiation measurement (ARM) user facility, a U.S. Department of Energy (DOE)
Office of Science user facility managed by the Office of Biological and Environmental Research.
The authors would like to specifically thank Allison Aiken (Los Alamos National Laboratory) for
useful discussions regarding the PASS-3 at the SGP site. The authors greatly acknowledge
Vanessa Selimovic, and Robert Yokelson from University of Montana for lending us the PAX-
405, Patrick Sheridan, John Ogren, and Derek Hageman from NOAA for lending us the CLAP,
and Anthony Prenni from the National Park Service for lending us the AETH during the FIREX
campaign. Nick Good (Colorado State University), Jim Roberts (NOAA) and Carsten Warneke
(NOAA) are acknowledged for their on-site support at FSL.





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
