# Peer review of "Development of a Universal Correction Algorithm for Use with any Filter-Based Absorption 1"

_Atmospheric Measurement Techniques, 2019_

## Referee Comment (RC1) · Anonymous Referee #1 · 31 Jan 2020

Li et al.: Development of a Universal Correction Algorithm for Filter-Based Absorption Photometers

Atmos. Meas. Tech. Discuss., doi.org/10.5194/amt-2019-336

**Review**

**General**

The paper presents a new algorithm for calculating absorption coefficient from CLAP, TAP, and PSAP data. It looks interesting and it may prove to be a very good one. The authors even give an Igor code for using it which is good, people can test it. The authors first evaluate older algorithms, the Bond et al. (1999) and the Virkkula et al. (2005) algorithms and then present their own and evaluate its performance. The data for the evaluations come from both lab-based and field measurements which is also good.

However, I found serious issues in the paper that should be corrected before the paper can be published in AMT. They are presented in the detailed comments below.

**Detailed comments**

- The algorithm is shown in Eq. (9) Looks interesting although quite complicated. How did you get AAE into it? Please show intermediate steps from Eqs. (8) to (9). I read the supplement but did not find steps leading to Eq. (9).

**Discussion on Fig 2 and Table 3.**

There are three strange things in your Fig. 2.

- If I compare the scatter plots in Fig. 2a of V2005 where the B1999 correction was used as such, with Fig. 3 in Virkkula (2010) where the B1999 correction with the coefficients updated by Ogren 2010 the regression lines don't change nearly as much as as in your Fig. 2. The difference between the original B1999 and that adjusted by Ogren (2010) is not big, definitely not as much as in your Fig. 2. and in Table 3. For instance, now you claim that the slope for the green changes from 2.5 to 1.01 when using the original B1999 formula and the one adjusted by Ogren (2010). That cannot be true. The additional correction factor Ogren (2010) derived was 0.97  $\pm$  0.01. Otherwise it is B1999. Also in your Table 3 the change of the slope from 2.83 to 1.03 when using either the original V2005 or the corrected constants in Virkkula (2010) is far from realistic. Both of these can easily be tested by using the constants from those papers. I have added some examples below.

- The second observation is that in the above-mentioned scatter plots in V2005 and Virkkula (2010) the absorption coefficients calculated with the B1999 correction without and with the updates are either lower than or close to the absorption standard. The V2005 correction and its adjustment in Virkkula (2010) resulted in increasing the absorption close to the 1:1 line of the respective scatter plots, not decreasing like in your Fig. (2). This is obvious especially for dark aerosol with SSA < 0.3 in those two papers.

- The third observation is that you get essentially the same absorption coefficients with B1999 and V2005 and their respective updates. In Fig 2 the data points are almost on top of each other. And their regression lines are almost identical. This would be correct for high SSA (>0.7) but now your SSA went down to < 0.6 for the FIREX data (your Fig 6) and < 0.5 in the SGP data (your Fig 9). For such low SSA the two methods do not yield so similar absorption coefficients unless you used only data where transmittance > 0.8. What Tr range was used for the regressions?

To evaluate the possibility of getting such strange relationships I present some calculations here.

By using the symbols of the paper the Bond et al. (1999) correction is

$$Babs = \frac{1}{C_2 Tr + C_3} Batn - C_1 Bscat$$

In Bond et al. (1999) Eq. (12) is

 $Babs = \frac{1}{1.22} \left( \frac{0.873}{1.079 \text{Tr} + 0.71} \text{Batn} - 0.02 \text{Bscat} \right) \approx \frac{1}{1.5087 \text{Tr} + 0.9922} \text{Batn} - 0.016 \text{Bscat}$

 $\Rightarrow$  C1  $\approx \! 0.016,$  C2  $\approx$  1.509, C3  $\approx 0.992$

Ogren (2010) added a very small adjustment factor, 0.97, so that the equation becomes

 $\begin{aligned} Babs = & \frac{1}{1.22} \left( \frac{0.97 \cdot 0.873}{1.079 \text{Tr} + 0.71} \text{Batn} - 0.02 \text{Bscat} \right) \approx \frac{1}{1.556 \text{Tr} + 1.023} \text{Batn} - 0.016 \text{Bscat} \\ \Rightarrow & \textbf{C}_1 \approx \textbf{0.016}, \, \textbf{C}_2 \approx \textbf{1.556}, \, \textbf{C}_3 \approx \textbf{1.023} \end{aligned}$

The algorithm in Virkkula et al. (2005) is of the form

 $B_{abs} = (C_4 + C_5(C_6 + C_7 SSA) In(Tr))B_{atn} - C_1 B_{scat}$

For 530 nm the constants were

 $C_4 = 0.306$ ,  $C_5 = -0.522$ ,  $C_6 = 1.234$ ,  $C_7 = -0.952$  and  $C_1 = 0.016$  in Virkkula et al. (2005) and  $C_4 = 0.358$ ,  $C_5 = -0.64$ ,  $C_6 = 1.17$ ,  $C_7 = -0.71$  and  $C_1 = 0.017$  in Virkkula (2010)

If we assume Batn = 500  $Mm^{-1}$ , Tr = 0.9 and Bscat = 100  $Mm^{-1}$  the absorption coefficient calculated from the orginal B1999, corrected with Ogren (2010), Virkkula et al. (2005), and the corrected Virkkula (2010) equations are ~211  $Mm^{-1}$ , 204  $Mm^{-1}$ , 176  $Mm^{-1}$  and 209  $Mm^{-1}$ , respectively.

At Tr = 0.5 the respective values are  $\sim$ 285 Mm-1, 276 Mm-1, 325 Mm-1, and 406 Mm-1.

Note that Babs calculated with the corrected Virkkula (2010) coefficients are close to the B1999 values only at high Tr. Note also that Babs calculated with the corrected Virkkula (2010) coefficients are higher than those calculated with the coefficients in the Virkkula et al. (2005). The ratio Babs(V2005)/Babs(V2010) is ~0.8 but it depends on Tr and SSA. But it is not the other way like in your Fig. 2.

So there is definitely some big mistake in the calculations. They have to be corrected for the revised manuscript. This means a large fraction of the paper has to be corrected because comparisons between your new method and the old ones are presented throughout the paper.

---

## Referee Comment (RC2) · Anonymous Referee #2 · 31 Jan 2020

The manuscript describes a new correction for filter-based absorption photometers. The method is compared with known correction methods. The authors explain that the method is "universal" due to the wide range of AAEs and SSAs. However, important factors known in the literature are not considered. Details will come later in the review.

The naming of this method in the title as "universal" method is wrong. It remains hidden to the reader that this method has only been developed and tested for specific aerosol types. The limitations are given in the conclusion. Furthermore, it is important to emphasise that the corrected factors for the Bond1999 and Virkkual2005 corrections are only valid for the specific aerosols investigated in this study. An application as a 'standardised' method, as suggested by the authors, is only possible after successful testing with different aerosol types.

Known correction methods are closely related to physical models. Terms of the corrections are associated with physical effects (e.g. load correction, scattering correction). The model presented here is based more on mathematical optimisation methods. A discussion of the derived parameters (G0 to G7) and how these parameters are related to aerosol properties is missing.

Detailed Review

Line 130: The authors point out the wide range of SSAs and AAEs. However, it is not shown to what extent these variables are correlated in the available data sets. Scatterplots of AAE, SSA and also SAE would provide valuable information. On this basis, an aerosol classification, as shown for example in Schmeisser et al. (2017), could give valuable information to what extent the data set is sufficient for a universal correction.

Chapter 2: Information on relative humidity or aerosol drying is missing. For developing correction methods, a chapter on instrument calibrations and description of corrections is required.

A consideration of the particle size effect is missing. It is known that smaller particles can penetrate deeper into the filter and absorb more light there by multiple scattering by the filter material. This effect was theoretically illustrated in Moteki et al (2010) and Nakayama et al (2010) and has been shown in laboratory measurements. According to Nakayama et al., this effect would explain differences of a factor of 2 and greater for mono disperse aerosol. This effect is unfortunately the worst quantified effect for complex aerosols. For the present manuscript this means that without a particle number size distribution of the particles containing absorbing material, the magnitude of the effect cannot be estimated. A correction is only possible with a complex aerosol physical characterization. Nevertheless, possible artefacts due to this effect should always be discussed.

The cross-sensitivity to scattering is important for high SSAs. The Magnitude depends

on particle size and wavelength (wavelength effect can be seen in Müller et al., 2011a). It was shown with radiative transfer calculations (Mueller et al., 2014) that this effect is linked to the scattering asymmetry parameter. In Mueller et al. (2014) it was also shown that for high SSAs (SSA>0.95) the corrections become strongly nonlinear and are only insufficiently described by current correction methods. This can also be found in equation 8 of the present manuscript. The second term in 'g' is (1-ssa)/(1-(1-c1)*ssa). In what range of SSAs equation 9 is a good approximation of equation 8?

The approach in equation 9 is a linear equation with interaction terms. According to radiative transfer calculations, this approach makes sense only in a limited range of parameter values . Why AAE is used as independent parameter is not quite obvious. Light scattering and absorption are independent for each wavelength. An indirect dependence on the aerosol type (e.g. composition) is possible. Unfortunately, the parameters and constant are only treated mathematically. A link to aerosol properties would be welcome. This could result in further boundary conditions for the mathematical fit and a more robust determination of the values of the constants might be possible.

In Li et al (2019) the FIREX data set has already been discussed under a different aspect. It was shown that PAX generally provides lower values than other instruments (TAP, CLAP, Offline, EC/OC, SP2). The authors wrote: "This could imply that most instruments over-estimate BC concentrations, but it could also imply that the PAX-870 measurement is incorrect. However, in the absence of a "ground truth" measurement, we cannot confirm or reject either claim. These discrepancies between instrument cannot be explained by measurement uncertainties alone." This gives rise to two possibilities: a) aerosol type or b) larger experimental uncertainties than expected from instrumental uncertainties

a) The special properties of the generated aerosol types are responsible for discrepancies during FIREX. Biomass burning is a challenge. Especially since the different types of instruments measure different properties! The PAX and CLAP/PSAP measure the light absorption and are therefore comparable in principle. It is correct to consider PAX

as a reference instrument, because filter based methods require complex corrections. The available data for the FIREX campaign must therefore be considered as being very specific and cannot be used as a universal calibration.

b) The discrepancies between PAX and PSAP/CLAP (see Fig. 5 in Li et al. 2019) for "Black" cluster like particles are significantly higher for the FIREX experiment than values from the literature. These differences require a precise presentation of the calibration, data processing and possible cross-comparisons to underline the high quality of data. Differences due to artefact of measurement techniques should be discussed in detail.

Line 198: From which instrument was the scattering Angström exponent (SAE) calculated? Where scattering data corrected for truncation?

The results of the multidimensional fit must be checked for physical meaning. Example Table S7, B1999-SGP-CLAP: The parameters C2 and C3 add up to about 5 for all wavelengths. This represents the correction factor for a fresh filter (Tr=1) and black aerosol ( negligible scattering). The strong wavelength dependence of C2 and C3, however, contradicts the literature. The results in Virkkula et al (2005) (see Figure 2) do not show such a strong wavelength dependence of calibration constants.

Line 306: Aim 1: The range of values of Babs is not important for any correction. Corrections are function of the optical depth of loaded particles (shown in Mueller et al (2014)). Concentration levels is merely important for performing experiments in an acceptable time and proper signal to noise.

Figure 3: Is the SSA given for the respective wavelength or for the wavelength of the reference device, the PAX?

Line 670: In Mueller et al. (2011a) no photoacoustic photometer was used. The "reference" instrument was a MAAP. In Arnott et al. (2005) a slope of 1.6 was determined. The measurements were, to the knowledge of the Reviewer, also performed with "SGP-

aerosol". Thus, the aerosol could have similar properties to the SGP measurements performed in this study and would not be a true independent measurement (speculation of the reviewer). Davis et al (2019) compared the B1999, V2005 and M2014 (Mueller et al. 2014) methods. Corrected values with the B1999 and V2005 methods were 40 to 50% higher than values of a PAS. The M2014 method could almost halve the deviations, although the M2014 method is identical to the B1999 and V2005 methods for low SSAs. The improvements are therefore clearly due to the improved scattering correction and probably also to the introduction of a mixed term between scattering and absorption. It would be very interesting to see how the performance of the M2014 method would be for the present FIREX measurements.

Line 701: The authors wrote "It is unclear how the algorithms will work for other absorbing aerosols (e.g., dominated by fossil fuel emissions or mineral dust)." There is a simple test. For an unloaded filter (ln(TR)=0) and assuming an aerosol with SSA=0.9 and AAE=1.1 , a not atypical aerosol, g can be calculated using the parameters listed in Table 4. Calculated values are: g=0.19 for FIREX/CLAP_green and g=-0.3 for SGP/CLAP_green. Negative values are not possible. What has the reviewer done wrong?

As already indicated, a deeper discussion of the results with regard to aerosol physical properties would be welcome. This could lead to an outlook for further correction models that include the advantages of the individual models.

The reviewer has not addressed the application of the new correction to Aethalometers. The reason is that the reviewer is in complete agreement with the authors that there should be a standard method, since the physics in PSAP, CLAP and Aethalometers is the same.

Literature

Schmeisser, L., Andrews, E., Ogren, J. A., Sheridan, P., Jefferson, A., Sharma, S., Kim, J. E., Sherman, J. P., Sorribas, M., Kalapov, I., Arsov, T., Angelov, C., Mayol-Bracero, O.

[Figure]

L., Labuschagne, C., Kim, S.-W., Hoffer, A., Lin, N.-H., Chia, H.-P., Bergin, M., Sun, J., Liu, P., and Wu, H.: Classifying aerosol type using in situ surface spectral aerosol optical properties, Atmos. Chem. Phys., 17, 12097–12120, https://doi.org/10.5194/acp-17-12097-2017, 2017.

Nakayama, T., et al. (2010). "Size-dependent correction factors for absorption measurements using filter-based photometers: PSAP and COSMOS." Journal of Aerosol Science 41(4): 333-343.

Moteki, N., et al. (2010). "Radiative transfer modeling of filter-based measurements of light absorption by particles: Importance of particle size dependent penetration depth." Journal of Aerosol Science 41(4): 401-412.

---

## Author Comment (AC1) · 27 Feb 2020

**Development of a Universal Correction Algorithm for Use with any Filter-Based Absorption Photometers (amt-2019-336)**

Reply to Anonymous Referee #1:

We appreciate the reviewer's time in reviewing our submission and are grateful for the comments and questions the reviewer provided. These appear to be related to some confusion that the reviewer had in reading our manuscript, which highlights to us that we were not as clear as we should have been with some of the text. We apologize for this lack of clarity and provide clarification in our response to the specific comments below.

*- The algorithm is shown in Eq. (9) Looks interesting although quite complicated. How did you get AAE into it? Please show intermediate steps from Eqs. (8) to (9). I read the supplement but did not find steps leading to Eq. (9).*

There are no intermediate steps between Equations 8 and 9, and in re-reading our text, we understand why the reviewer has this expectation.

Equation 8 represents a generalized form of both the Bond and Virkkula correction equations, i.e., for either formulation of $f(Tr(\lambda))$ present in the literature. Equation 9 is our new proposed correction equation for the "g term" in Equation 8, which was developed as a multiple linear regression using $\ln(Tr)$, SSA, and AAE with interaction terms. To clarify, we have modified the text in Line 345 to read "We define a *new* function "g" that can be used in Eq. (8). Specifically, we construct a multivariate linear model for "g", introducing AAE as a dependent variable and including interaction terms between SSA, AAE, and $\ln(Tr)$… "

*- The remainder of the reviewer's comments are related to Bond et al. (1999), which was updated in Ogren (2010), and Virkkula et al. (2005), updated by Virkkula (2010).*

First, we would like to clarify that we refer to as "updated B1999" and "updated V2005" are not simply the updated corrections in Ogren (2010) and Virkkula (2010). Those updated corrections are what we refer to as "B1999" and "V2005" in e.g., Figure 2. For our "updated B1999" and "updated V2005" results, we have re-fit the coefficients using the respective "g functions" from those papers, yielding a new set of coefficients that we provided as Table S7. We have added a new section (Section 2.4.3) to indicate that our updates are based on new coefficients rather than the adjustments from Ogren (2010) and Virkkula (2010).

Line 316: "2.4.3 Refitting the coefficients in B1999 and V2005

With the reference measurements of $B_{abs}$ from the photoacoustic instruments, we are able to refit the coefficients in the B1999 and V2005 corrections ($C_2$ to $C_7$ in Eq. (5) and Eq. (6)) using our data. Specifically, we use the Levenberg-Marquardt algorithm (1944) to iteratively fit the coefficients until the chi-square of the coefficients are minimized. The fitting is implemented using the "Curvefit" function in Igor Pro. It is worth noting that the derived coefficients may only be valid for the SGP and FIREX data. For aerosol properties different from our study, the optimal coefficients are likely to be different from the ones reported here. Hereafter, the B1999 and V2005

results with refitted coefficients are referred to as "updated B1999" and "updated V2005", respectively."

This appears to be the root of the reviewer's concern, but we provide additional information below.

**Response to the reviewer's first and second observations on Figure 2 and Table 3:**

*- If I compare the scatter plots in Fig. 2a of V2005 where the B1999 correction was used as such, with Fig. 3 in Virkkula (2010) where the B1999 correction with the coefficients updated by Ogren 2010 the regression lines don't change nearly as much as as in your Fig. 2. The difference between the original B1999 and that adjusted by Ogren (2010) is not big, definitely not as much as in your Fig. 2. and in Table 3. For instance, now you claim that the slope for the green changes from 2.5 to 1.01 when using the original B1999 formula and the one adjusted by Ogren (2010). That cannot be true. The additional correction factor Ogren (2010) derived was 0.97 ± 0.01. Otherwise it is B1999. Also in your Table 3 the change of the slope from 2.83 to 1.03 when using either the original V2005 or the corrected constants in Virkkula (2010) is far from realistic. Both of these can easily be tested by using the constants from those papers. I have added some examples below.*

*The second observation is that in the above-mentioned scatter plots in V2005 and Virkkula (2010) the absorption coefficients calculated with the B1999 correction without and with the updates are either lower than or close to the absorption standard. The V2005 correction and its adjustment in Virkkula (2010) resulted in increasing the absorption close to the 1:1 line of the respective scatter plots, not decreasing like in your Fig. (2). This is obvious especially for dark aerosol with SSA < 0.3 in those two papers.*

Since Ogren (2010) and Virkkula et al. (2010) have respectively adjusted the original corrections (Bond et al. (1999) and Virkkula et al. ((2005))) to universal PSAP spot area and "true" flow rate, we expect the coefficients reported in their publications to be more precise. Thus, we simply use these coefficients when correcting our $B_{ATN}$ data (the coefficients are presented in Table 2), instead of the ones in the original publications. The coefficients provided by the reviewer at the end of the comment are same as the ones reported in Table 2.

Using the coefficients in Table 2 (B1999 and V2005) and Table S7 (B1999 and V2005 updated coefficients), we generated Figure 2 and Table 3. As $B_{abs}$ from the photoacoustic instruments was used as reference when updating the coefficients, the red and blue curves (B1999 and V2005 updated coefficients) are closer to 1:1 in Figure 2 and yield slopes closer to 1 in Table 3.

**Reply to the reviewer's third observation on Figure 2 and Table 3:**

*- The third observation is that you get essentially the same absorption coefficients with B1999 and V2005 and their respective updates. In Fig 2 the data points are almost on top of each other. And their regression lines are almost identical. This would be correct for high SSA (>0.7) but now your SSA went down to < 0.6 for the FIREX data (your Fig 6) and < 0.5 in the SGP data (your Fig 9). For such low SSA the two methods do not yield so similar absorption coefficients unless you used only data where transmittance > 0.8. What Tr range was used for the regressions?*

We definitely agree with the reviewer that the performance of B1999 and V2005 may vary with Tr and SSA. The simulated $B_{abs}/B_{atn}$ ("g" term in Eq. 8 and Eq. 9) using different values of Tr and SSA are presented below. The results of B1999 and V2005 are generated using the updated coefficients from Table S7 (FIREX-CLAP). As the reviewer mentioned, the B1999 and V2005 corrections are more agreed for greater values of Tr (as seen in panel d) and the combination of lower values of Tr and greater values of SSA (as seen in panels e and f).

[Figure]

**Figure 1.** Simulated "g" term (528 nm) in Eq. (8) or Eq. (9). In panel a) and b), the grey regions correspond to "g" values less than 0.16. The results of the B1999 and V2005 are generated using updated coefficients from Table S7 (FIREX-CLAP).

Regarding the reviewer's comment about the overlap in B1999 and V2005 in Figure 2, we provide the following figures to demonstrate our results. In these figures, $B_{abs}$ derived by the two corrections (467 nm as an example) are plotted against each other, and colored by either SSA (0.3-1) or Tr (0.4-1). Here, we zoom in the original axes (0 - 7000 Mm$^{-1}$) in Figure 2 to better display our results (0 - 2000 Mm$^{-1}$). As seen in the figures, the biases between the two corrections are apparently associated with SSA (yet less obvious for Tr), consistent with the trends observed in the previous figure.

[Figure]

**Figure 2.** $B_{abs}$ (B1999) against $B_{abs}$ (V2005) for the FIREX data (467 nm). The points are colored by a) SSA and b) Tr. In panel b), there are a few black points, which are associated with $0.25 <$ Tr $< 0.4$.

We apologize again for some of the ambiguity in our original manuscript regarding B1999, V2005, and our updated versions of both with new coefficients. The revisions prompted by the reviewer's comments have resulted in more clarity within the manuscript, and we thank the reviewer again for his/her comments.

---

## Author Comment (AC2) · 27 Feb 2020

**Development of a Universal Correction Algorithm for use with any Filter-Based Absorption Photometers (amt-2019-336)**

Reply to Anonymous Referee #2:

We thank the reviewer for the careful reading of our manuscript and for providing a set of very thoughtful comments and critiques, which will be addressed in turn below. Below we restate your editorial comments in italics, and then indicate our response and revision. Please note that all the revised text and references in this response document have been included in the revised manuscript.

The manuscript describes a new correction for filter-based absorption photometers. The method is compared with known correction methods. The authors explain that the method is "universal" due to the wide range of AAEs and SSAs. However, important factors known in the literature are not considered. Details will come later in the review.

**Response #1:** We first would like to kindly clarify that our intent in calling this a "universal" algorithm is due to its potential application on different filter-based absorption photometers (including PSAP, CLAP, TAP and Aethalometer), rather than its applicability to a wide range of AAE and SSA. The reviewer appears to have capture this intent with her/his final review comment: "*The reviewer has not addressed the application of the new correction to Aethalometers. The reason is that the reviewer is in complete agreement with the authors that there should be a standard method, since the physics in PSAP, CLAP and Aethalometers is the same,"* but this appears to be a different point entirely.

We did try to emphasize our intent for a "universal" algorithm in our original submission, e.g.:

Introduction:

Line 114: In addition, we propose "universal" correction algorithms that are applicable to any filter-based absorption photometer (e.g., CLAP, TAP, PSAP, and AETH) across multiple wavelengths by combining observed filter-based  $B_{abs}$  with  $B_{scat}$  (e.g., from a co-located nephelometer (NEPH)) and reference  $B_{abs}$  (e.g., from a co-located photoacoustic instrument).

Line 136: Our algorithms were then extended to the AETH data from the FIREX laboratory campaign and the PSAP data collected at the SGP site to verify the "universal" nature of the algorithms.

Discussion:

Line 660: Overall, the derived properties using the new correction are consistent across all instruments, suggesting its universality.

**Conclusion:**

Line 728: We also developed "universal" correction algorithms that are applicable to any filterbased absorption photometer (e.g., PSAP, CLAP, TAP, AETH), which will have utility for any historic or future filter-based absorption measurements and which have the potential to standardize absorption coefficients across all filter-based instruments. However, this critical disconnect between our intent and the reviewer's interpretation of our intent exists. To clarify for the reviewer and any other reader of this work, we have added and/or revised text as follows:

We revised the title of our manuscript to "Development of a Universal Correction Algorithm **for Use with any** Filter-Based Absorption Photometer".

Line 118 (new text): "We emphasize that the "universal" feature of our algorithm is based on its applicability to different filter-based absorption photometers, rather than the ranges of aerosol parameter tested in this work."

line 337 (new text): "..., but we have the additional aim of developing a model that is applicable to any filter-based absorption photometer."

Line 562 (revised text, bolded here for emphasis): "To explore the universal behavior of the new algorithms across different instruments, we next apply our algorithms to the other filterbased absorption photometers operated during the FIREX study (TAP and AETH)."

The naming of this method in the title as "universal" method is wrong. It remains hidden to the reader that this method has only been developed and tested for specific aerosol types. The limitations are given in the conclusion. Furthermore, it is important to emphasise that the corrected factors for the Bond1999 and Virkkual2005 corrections are only valid for the specific aerosols investigated in this study. An application as a 'standardised' method, as suggested by the authors, is only possible after successful testing with different aerosol types.

**Response #2:** As we stated above, the so-called "universality" is not based on aerosol properties, so we again apologize for this confusion. Regardless, we do agree with the reviewer that our correction factors derived by B1999 and V2005 are only valid for the aerosols tested in this work. Therefore, we added the following sentence to emphasize this in Line 422: "It is important to keep in mind that the updated coefficient values of B1999 and V2005 (Table S7) are only valid for the aerosols investigated in this study. Future experiments are needed to systematically determine how the coefficients in B1999 and V2005 may change for different aerosol types."

Moreover, we added a new section (section 2.4.3) describing how we updated the coefficients in B1999 and V2005.

Line 316: "2.4.3 Refitting the coefficients in B1999 and V2005

With the reference measurements of  $B_{abs}$  from the photoacoustic instruments, we are able to refit the coefficients in the B1999 and V2005 corrections (C2 to C7 in Eq. (5) and Eq. (6)) using our data. Specifically, we use the Levenberg-Marquardt algorithm (1944) to iteratively fit the coefficients until the chi-square of the coefficients are minimized. The fitting is implemented using the "Curvefit" function in Igor Pro. It is worth noting that the derived coefficients may only be valid for the SGP and FIREX data. For aerosol properties different from our study, the optimal coefficients are likely to be different from the ones reported here. Hereafter, the B1999 and V2005 results with refitted coefficients are referred to as "updated B1999" and "updated V2005", respectively."

Known correction methods are closely related to physical models. Terms of the corrections are associated with physical effects (e.g. load correction, scattering correction). The model presented here is based more on mathematical optimisation methods. A discussion of the derived parameters (G0 to G7) and how these parameters are related to aerosol properties is missing.

**Response #3**: We first would also like to correct an error in Eq. 9 which the reviewer highlights by the calculation of g < 0 in a later comment: the parameters after  $G_5$  and  $G_6$  ( $ln(\lambda) \times AAE$  and SSA ( $\lambda$ ) × AAE, respectively) should be exchanged. Eq. 9 should be:

 $g(Tr(\lambda), SSA(\lambda), AAE) = G_0 + G_1 \times ln(Tr(\lambda)) + G_2 \times SSA(\lambda) + G_3 \times AAE + G_4 \times ln(Tr(\lambda)) \times SSA(\lambda) + G_5 \times SSA(\lambda) \times AAE + G_6 \times ln(Tr) \times AAE + G_7 \times SSA(\lambda) \times AAE \times ln(Tr(\lambda))$

The coefficient values in Table 4 and in our supplemental computer code were derived using this correct form of the equation, and there was simply an error in this equation in the manuscript text. We apologize to the reviewer for causing this confusion and are very grateful to her/him for bringing this to our attention. (This explains the negative "g values" that the reviewer discusses in a later comment).

Towards this comment, in the original manuscript, the development and interpretation of Eq. 9 were included in the supplementary materials (pages 2-6), as subsections of "model development". In the main text, we briefly interpreted how these parameters relate to SSA, AAE, and Tr (Line 350). To further relate these parameters with aerosol properties, we add the following sentences:

Line 352: "Although Eq. (9) is developed based on statistical approaches, we attempt to relate this statistical model to physical effects. The coefficients  $G_1 - G_3$  are fairly straightforward, as these account for the influence of filter loading (G1), relative light scattering by the aerosols (G2), and the brownness of the aerosols (G3). The interaction terms (G4 - G7) are more difficult in assigning a physical meaning; however, the interaction between filter loading and relative light scattering (G4) is might be interpreted as an absolute light scattering by the aerosols on the filter, while the interaction between filter loading and aerosol brownness (G6) is somewhat analogous to G4. The three-way interaction between filter loading, scattering and brownness of aerosols (G7) is required because of the three two-way interaction terms.

To further this physical interpretation of our statistical model (Eq. 9), we explore the relationship between  $\frac{B_{abs}}{B_{ATN}}$  and ln(Tr), which essentially follows a "y = m·x + b" form, where y is  $\frac{B_{abs}}{B_{ATN}}$  and x is ln(Tr). The slope (m) is defined as G1 + G4×SSA + G6×AAE + G7×SSA×AAE, and the intercept (b) is defined as G2×SSA + G3×AAE + G5×SSA×AAE. Therefore, different combinations of SSA and AAE modulate this relationship between  $\frac{B_{abs}}{B_{ATN}}$  and ln(Tr). For example, loading "black" particles on the filter (e.g., AAE ~1 and SSA ~0.3) tends to produce larger values of  $\frac{B_{abs}}{B_{ATN}}$ , while loading "white" particles on the filter (e.g., AAE ~3 and SSA ~0.9) tends to produce smaller values of  $\frac{B_{abs}}{B_{ATN}}$  (see Fig. S3 of this work and Fig. 4 in Virkkula et al. (2005)). This relationship becomes more complex when considering, e.g., mixed sulfate and black carbon particles; SSA can be high while AAE is low, and the corresponding  $\frac{B_{abs}}{B_{ATN}}$  can be variable (also see Fig. S3). Therefore, in order to properly compensate for the effects of loading and aerosol optical properties, a multiple linear regression with interaction terms is introduced in Eq. (9)."

**Detailed Review**

Line 130: The authors point out the wide range of SSAs and AAEs. However, it is not shown to what extent these variables are correlated in the available data sets. Scatterplots of AAE, SSA and also SAE would provide valuable information. On this basis, an aerosol classification, as shown for example in Schmeisser et al. (2017), could give valuable information to what extent the data set is sufficient for a universal correction.

**Response #4**: Thank you for this discerning comment. It was our hope that the use of a broader range of SSA and AAE could make this more widely applicable, but as the reviewer has pointed out elsewhere, this will not be strictly true (and we agree!). Regardless, to address this comment, we generated Figure R1 for the SGP data using similar "AAE-SAE space" as that in Cappa et al. (2016) and Schmeisser et al. (2017). In panel a), our data are colored by SSA and the SGP results from Schmeisser et al. (2017) is illustrated by the brown marker and error bars. In panel b), the AAE and SAE results of all NOAA/ESRL sites from Schmeisser et al. (2017) overlap with our SGP data.

Here, we first emphasize that the SGP data used in our work are subsamples of the campaign reported in Schmeisser et al. (2017), but in general, the AAE and SAE values calculated for the SGP site in our work agree well with the values reported for SGP in Schmeisser et al. (2017). Moreover, our results of AAE and SAE overlap with most of the values from the NOAA/ESRL sites, except when marine aerosols or dust contribute to the local aerosol populations. Though none of the NOAA/ESRL sites fall into the clusters of BC or BC/BrC, some of our data can represent the optical properties of aerosols from these clusters. Therefore, we highlight that our algorithm developed by the SGP data *may* have the potential to be generalized to a variety of environmental conditions, but we would need to validate this using observations from more studies. We again emphasize that our intent was to develop an algorithm that is universal across all filter-based absorption photometers, and not universal across all aerosol types, so such a validation is outside the scope of this work.

**Figure R1.** AAE vs. SAE for the SGP data. The three parameters are calculated using photoacoustic  $B_{abs}$  and Nephelometer  $B_{scat}$ . The panels are overlaid with the classification scheme presented in Cappa et al. (2016) and Schmeisser et al. (2017). In panel a), the averaged values (and standard deviation) of AAE and SAE reported for the SGP site in Schmeisser et al. (2017) are illustrated by the brown marker and error bars. Our results are colored by the corresponding SSA. In panel b), the results in Schmeisser et al. (2017) are colored by the types of observational station and our results are colored in grey.

Figure R1 has been included in the revised supplementary material (Figure S15) and revision was made accordingly:

Line 252: "The scatterplot of AAE and SAE for the SGP data can be found in Fig. S15. Our results of AAE and SAE are compared to the values reported for different NOAA Earth System Research Laboratory (ESRL) observational sites in Schmeisser et al. (2017)".

Line 608: "To investigate if our algorithms are suitable to correct  $B_{abs}$  obtained from different ambient environments, the aerosol properties from the SGP site are compared to those from the other NOAA/ESRL observational sites. We use the similar "AAE-SAE space" as that in Cappa et al. (2016) and Schmeisser et al. (2017) to infer dominant aerosol types (e.g., BC, BrC, dust, mixed dust/BC/BrC, see Fig. S15). ..."

To further investigate how different algorithms apply to different aerosol properties, we generated Figures R2 and R3 (SGP and FIREX, respectively), in which the variable on y axis is  $B_{abs}$  ratio = corrected  $B_{abs}$  (different corrections) /  $B_{abs}$  from photoacoustic instruments (reference), and the parameters on x axis include relative humidity (RH), AAE, SAE, and SSA (528 nm). In general, an apparent association between the  $B_{abs}$  ratio and these parameters exists in the uncorrected data (raw  $B_{ATN}$ ), and this association persists when using B1999 and V2005, especially for RH and SSA. However, these associations are reduced or eliminated when applying our algorithm on the filter-based absorption measurements. Although RH and SAE are not included in our algorithm, our algorithm appears to account for any influence that these parameters have on the measurements. This effect may be captured by one of the interaction terms in our statistical model, but as discussed in Response #3 above, it is difficult to assign physical meaning to these. These figures will also

support our response to some of the reviewer's comments later in this file (Response #5 and Response #6).

Revision was made accordingly:

Line 375: "We evaluate the model by plotting  $\frac{B_{abs}}{B_{ATN}}$  against aerosol properties not included in Eq. (9) (such as relative humidity and aerosol geometric mean diameter, which have been previously reported to bias corrections of filter-based Babs, (Moteki et al., 2010; Nakayama et al., 2010; Schmid et al., 2006)). The results are presented in Fig. S5-S7."